# Effects of Transient Processes for Thermal Simulations of the Central European Basin

Denise Degen[1] and Mauro Cacace[2]

[1]Computational Geoscience and Reservoir Engineering (CGRE), RWTH Aachen University, Wüllnerstraße 2, 52072 Aachen, Germany

[2]Helmholtz Centre Potsdam GFZ German Research Centre for Geosciences, Telegrafenberg, 14473 Potsdam, Germany

**Correspondence:** Mauro Cacace (mauro.cacace@gfz-potsdam.de)

**Abstract.** Transient processes play a major role in geophysical applications. In this paper, we quantify the significant influence arising from transient processes for conductive heat transfer problems for sedimentary basin systems. We demonstrate how the thermal properties are affected when changing the system from a stationary to a non-stationary (transient) state and what impact time-dependent boundary conditions (as derived from paleoclimate information) have on the system's overall response.
Furthermore, we emphasize the importance of the time-stepping approach adopted to numerically solve for the transient case and the overall simulation duration since both factors exert a direct influence on the sensitivities of the thermal properties. We employ global sensitivity analyses to quantify not only the impact arising from the thermal properties but also their parameter correlations. Furthermore, we showcase how the results of such sensitivity analysis can be used to gain further insights into the complex Central European Basin System, in central and northern Europe. This computationally very demanding workflow becomes feasible through the construction of high precision surrogate models based on the reduced basis (RB) method.

## 1 Introduction

A proper quantification of the thermal state of sedimentary basins is of primary interest for subsurface exploration studies being especially relevant in relation to ongoing systematic efforts that are made worldwide to develop concepts of proof for low-carbon energetic solutions. Among others, geothermal resources buried in the underground of sedimentary basins and/or in volcanic areas are increasingly harvested either for direct heating usage or for electricity generation (Fridleifsson, 2001; Fridleifsson et al., 2008; Glassley, 2014). However, the development of geothermal projects requires extensive and site-specific studies of the underground thermal regime, which can only be predicted within a certain degree of confidence due to limitations in available observations. Heat flow measurements, temperature logs, and thermochronological data provide the basic observations to characterize the evolution and spatial distribution of temperature in the underground. However, these datasets are generally sparse and lacking in coverage to provide enough information for a proper assessment of available geothermal resources (Horváth et al., 2015; Schellschmidt et al., 2002). An alternative is to rely on process-oriented mathematical models that incorporate the details of the subsurface geology and the driving physics responsible for the observations done in the field. On the scale of the whole lithosphere, heat conduction is the main heat transport mechanism. The effects of fluid mediated processes are usually less relevant if not locally. The regional thermal configuration of a conductive lithosphere reflects to a

first-order the available heat in place. The latter depends on the regional tectonothermal configuration, which evolved through geological timesdue to:

    – varying thermal loading conditions as provided by the underlying convective mantle,

    – the amount of heat generated by dissipative underground processes (and therefore on the local geology), and,

    – lastly, the (time-varying) surficial climate conditions (Turcotte and Schubert, 2002).

It has been long recognized that the near-surface temperature distribution can maintain a "thermal memory" of the past surface boundary conditions. If conduction is considered as the only active heat transport mechanism, a variation in surface temperature propagates downward with a signal attenuation that scales with the square root of the internal period times the thermal diffusivity of the plate (Turcotte and Schubert, 2002). Given common ranges of thermal rock properties, daily and annual surface temperature variations are damped down to a depth of few tens of meters and therefore believed not to affect the tempera-

ture at greater depths. The situation changes when considering long-term variations in surface temperature as occurring over a glacial cycle. This could potentially affect the temperature gradient down to significant depths (kilometers scale) (Turcotte and Schubert, 2002). Despite these observations, commonly, studies of the thermal state of the continental lithosphere consider steady-state conditions, the working assumption being that of instantaneous thermodynamic equilibrium under a spatially variable but constant in time set of loading conditions (Bayer et al., 1997; Noack et al., 2012; Freymark et al., 2017; Fuchs

and Balling, 2016; Scheck-Wenderoth and Maystrenko, 2013). Transient effects are generally considered to be of secondary relevance and as such have received so far little attention. These effects are due to fluid mediated processes and, as relevant for the current study, to long term surface temperature variations (Ebigbo et al., 2016; Freymark et al., 2019; Mottaghy et al., 2011; Noack et al., 2013). Corrections to a steady-state geotherm for paleoclimatic effects require to account for time-varying surface boundary conditions. Such boundary conditions can be derived from available Earth System Models (ESM hereafter). This

requires (i) an efficient transfer of information from a global to a (sub)regional resolution as typical for subsurface geothermal studies, and (ii) an analysis of the sensitivity of the parameters at play (i.e. rock thermal properties) within proper confidence intervals. Under steady-state conditions, model validation is generally achieved by manual "tuning" of the rock parameters (thermal conductivity and heat production) within specified ranges. However, the dimension of the parameter space for a transient system poses serious computational limitations. This aspect can explain why the sensitivity of transient processes on the

regional thermal characteristics has never been neither investigated nor quantified.

To overcome these problems, we here demonstrate how to properly quantify the thermal state of a conductive lithosphere, including an in-depth and deterministic consideration of the sensitivity of the parameters at play as they can vary within proper confidence intervals. We will describe and discuss an automated, software-based workflow to achieve this goal, which also enables us to take into account transient boundary effects as derived from paleoclimate reconstruction studies. Based on the

developed workflow, we will demonstrate the relevance of such transients on the overall parameter sensitivity when compared to an analysis done under the assumption of steady-state thermal equilibrium.

The need to consider paleoclimate effects has been long recognized. However, so far, it has only been considered for correction

of (1D) vertical temperature gradients (Clauser, 1984; Majorowicz et al., 2008; Gosnold et al., 2011; Westaway and Younger, 2013; Dentzer et al., 2016) or the surface heat flow (Majorowicz and Wybraniec, 2011). Its influence on the calibration of thermal properties, such as thermal conductivity and radiogenic heat production, has never been investigated. Therefore, the study aims to investigate the sensitivity of the regional thermal characteristic of a lithospheric plate while moving away from a stationary state representation. The approach enables then to quantify the effects of paleoclimate conditions on the current thermal state of sedimentary basins, with a special focus on their influences on the calibration of rock thermal properties.

Our choice of a global sensitivity analysis stems from the results of Degen et al. (2020a), who demonstrated how a local sensitivity analysis likely leads to overestimating the influence of the model properties on the same model response. For this reason, we present a global sensitivity analysis to determine to which degree the thermal properties are impacted by considering transient processes. Additionally, we investigate if also the parameter correlations are affected by considering different physical processes.

Given the high computation demands of a global sensitivity analysis, we hereby rely on surrogate models. Hence, we use the reduced basis (RB) method to construct our surrogate model. The RB method is a Model Order Reduction (MOR) technique that aims at significantly reducing the spatial and temporal degrees of freedom of, as applied in this study, finite element problem formulations. The RB method has been widely studied by, for example, Grepl and Patera (2005); Hesthaven et al. (2016); Prud'homme et al. (2002); Quarteroni et al. (2015) for mathematical benchmark examples, and for the first time by Degen et al. (2020b) in a geoscientific context. In contrast to other statistical methods including Kriging and response surfaces (Baş and Boyacı, 2007; Bezerra et al., 2008; Frangos et al., 2010; Khuri and Mukhopadhyay, 2010; Miao et al., 2019; Mo et al., 2019; Myers et al., 2016; Navarro et al., 2018), the RB method enables the retrieval of the entire state variable (i.e. temperature). Thus, we make use of the RB method to guide the construction of the surrogate model.

In the present study we use the Central European Basin System (CEBS) in northern and central Europe as our study case (natural laboratory) (Maystrenko et al., 2013; Scheck-Wenderoth and Maystrenko, 2013; Scheck-Wenderoth et al., 2014). Our choice stems from the fact that the CEBS (i) represents a rather complex intracontinental basin, thereby representing a proper test case for our novel approach, as well as (ii) it is an area of interest for both past hydrocarbons and, currently low-to-middle enthalpy geothermal exploration.

The paper is structured as followed: In Section 2, we introduce the concepts of the global sensitivity study together with the main governing equations solved for and the paleotemperature data used as boundary conditions. The results of the steady-state analyses, the influence of transient boundary conditions, and transient processes are described in Section 3. This is followed by the discussion of the results in Section 4 and a final conclusion in Section 5.

## 2   Materials and Methods

In the following, we introduce the methodology of the sensitivity analyses used throughout this paper. The reader is referred to previous works by (Sobol, 2001; Saltelli, 2002; Saltelli et al., 2010) for details on global sensitivity analysis, while the study by Wainwright et al. (2014) provided an in-depth comparison of local and global sensitivity analyses. The applicability and

benefits of global sensitivity analyses applied to basin-scale thermal models have been discussed in details in a previous study by (Degen et al., 2020a).

## 2.1 Global Sensitivity Analysis

A sensitivity analysis aims to determine the influence of the model parameters (for our study, thermal parameters) on the model response (temperature in our case). We employ a global variance-based sensitivity analysis, namely the Sobol sensitivity analysis. In contrast, to a local sensitivity analysis, the Sobol method investigates the entire parameter domain. Based on the variances, sensitivity indices are derived, defining the influence of the parameters themselves and their correlations (Sobol, 2001). To reduce the number of forward solves required for the calculation of the indices, we use the Saltelli sampling routine (Saltelli, 2002; Saltelli et al., 2010).

The global sensitivity analysis is done by relying on the SALib Python library (Herman and Usher, 2017). To avoid statistical errors, we use 100,000 realizations per parameter for the steady-state and 10,000 for the transient analyses. Statistical errors can be, for instance, higher first-order than total-order contributions. This is physically not plausible since the total-order contributions compromise the influence of the parameters themselves plus the parameter correlation, whereas the first-order contributions compromise only the influence of the parameters themselves. For further information regarding the global sensitivity analysis, we refer to Sobol (2001) and for the sampling routine to Saltelli (2002); Saltelli et al. (2010). A comparison between local and global sensitivity analyses is provided in Wainwright et al. (2014) for hydrological models and in Degen et al. (2020a) for a basin-scale geothermal model.

As the quantity of interest, we hereby define the total amount of heat available in the model (steady-state analyses), and the total amount of heat available in the model over all time steps considered (transient analyses). Our choice enables us to quantify the influence of the paleotemperatures on the physical processes being investigated. At this stage, it is worth mentioning that the focus of the current study paper is not to provide an overall fit to available measurements. This is why we do not use, as commonly done, the misfit between measurements and simulated temperatures as our quantity of interest. This is also the main reason behind our choice to employ the RB method for the surrogate model construction. Other methods such as Kriging and response surfaces (Baş and Boyacı, 2007; Bezerra et al., 2008; Frangos et al., 2010; Khuri and Mukhopadhyay, 2010; Miao et al., 2019; Mo et al., 2019; Myers et al., 2016; Navarro et al., 2018) construct surrogate models for the observation space only. This entails that every value outside this space has to be obtained via inter- and extrapolation routines. Therefore, these approaches share the disadvantage of not taking the physical laws describing the process of interest into consideration. These limitations would have severely impacted the current study that focuses on understanding the effects of the physical processes of heat transport on the resulting rock properties and thermal state of the lithosphere rather than on a crude fit of temperature values at certain measurement locations. In other words, our interest here is in the entire temperate state of the plate. Thereby, our surrogate model proves useful since it is by definition physics-preserving. This to say that we can retrieve temperature values at every location in the model and, thus can determine the relevance of all thermal properties as relevant for the physics at play, steady/transient heat conduction.

## 2.2 Forward Problem

In order to improve the efficiency of the solvers and to investigate the relative importance of rock thermal properties on the resulting thermal configuration, we make use of adimensional forms of all relevant equations in this study.

In this paper, we consider both steady-state and transient conductive heat transfer simulations. For the steady-state conductive heat transfer, we take the radiogenic heat production as the source term into account (Turcotte and Schubert, 2002). Following the derivation presented in Degen et al. (2020a), we obtain:

$$\frac{\lambda}{\lambda_{\text{ref}}\, S_{\text{ref}}} \frac{\nabla^2}{l_{\text{ref}}^2} \left(\frac{T - T_{\text{ref}}}{T_{\text{ref}}}\right) + \frac{S}{S_{\text{ref}}\, T_{\text{ref}}\, \lambda_{\text{ref}}} = 0, \tag{1}$$

where $\lambda$ is the rock thermal conductivity, $T$ the temperature, $S$ the radiogenic heat production, $\lambda_{\text{ref}}$ the reference thermal conductivity, $T_{\text{ref}}$ the reference temperature, and $S_{\text{ref}}$ the reference radiogenic heat production.

Following a similar procedure, we can derive the following adimensional partial differential equation (PDE) for the transient case:

$$\frac{\alpha}{\alpha_{\text{ref}}\, S_{s,\text{ref}}} \frac{\nabla^2}{l_{\text{ref}}^2} \left(\frac{T - T_{\text{ref}}}{T_{\text{ref}}}\right) + \frac{S_s}{S_{s,\text{ref}}\, T_{\text{ref}}\, \alpha_{\text{ref}}} = \frac{\partial \frac{T - T_{\text{red}}}{T_{\text{ref}}}}{\partial \frac{t}{t_{\text{ref}}}}, \qquad \text{with } t_{\text{ref}} = \frac{1}{S_{s,\text{ref}}\, \alpha_{\text{ref}}}, S_s = \frac{S}{\rho c_p} \tag{2}$$

Here, $\alpha$ is the thermal diffusivity , $t$ time, $\alpha_{\text{ref}}$ the reference thermal diffusivity, $t_{\text{ref}}$ reference time, $\rho$ the density, $c_p$ the specific heat capacity, $l_{\text{ref}}^2$ the reference length, and $S_{s,\text{ref}}$ the reference value for the radiogenic heat production divided by the product of density and specific heat capacity. Note that both in Eq. 1 and 2 the Laplace operator acts on the normalized space. To close the system of equations 1 or 2 requires proper boundary conditions. Throughout the entire paper, we apply both at the upper and lower model boundaries Dirichlet-type first-order boundary conditions. The lower boundary condition corresponds to the 1300 °C isotherm (Turcotte and Schubert, 2002). Values imposed along the upper boundary differ for each analysis and will be discussed in the respective sections.

### 2.2.1 Surrogate Model Construction

The surrogate model used in this study is based on the RB method. RB is a model order reduction method, aiming at finding low dimensional representations for the high dimensional finite element simulations, as considered in this study. For an inverse process, such as a global sensitivity study, we need to perform several forward simulations by varying relevant rock properties. Due to the high number of forward solves, the problem, therefore, becomes computationally too demanding if relying on the high fidelity finite element forward simulation. Thereby the idea to "train" a model that is representative of a pre-defined range of rock properties. This trained model is a low dimensional representation of our original finite element problem. Note that we simplify the model in the mathematical instead of the physical domain. This means that we avoid introducing errors through, for instance, simplifying the physics or considering a smaller spatial and or temporal domain.

The RB method is by definition subdivided into an offline and an online phase. During the offline phase, performed only once per study, the surrogate model is constructed. All expensive computations take place during this stage only. Based on this offline stage, we can then use the developed surrogate model in outer loop processes, such as global sensitivity analysis. All

simulations that make use of the reduced model are part of what is referred to as the online stage. Further information regarding the RB method can be found in (Hesthaven et al., 2016; Prud'homme et al., 2002).

We generate all reduced models with the software package DwarfElephant (Degen et al., 2020b). DwarfElephant is based on the Multiphysics Object-Oriented Simulation Environment (MOOSE), a state-of-the-art finite element solver primarily developed by the Idaho National Laboratory (Alger et al., 2019). The setup and construction of the reduced model are analog to the one described in Degen et al. (2020a) and here omitted for the sake of clarity.

## 2.3 Paleoclimate Boundary Condition

The paleotemperature data (Fig. 1) that we use as an input for our transient boundary condition investigation has been obtained from the Max-Planck-Institute Earth System Model (MPI-ESM) (Giorgetta et al., 2013a). The MPI-ESM considers the exchange of energy, momentum, water, and carbon dioxide to couple the atmosphere, the ocean, and the land surface. For the atmosphere it is based on ECHAM6 (Stevens et al., 2013), for the ocean on MPIOM (Jungclaus et al., 2013), for the ocean's biogeochemistry on HAMOCC (Ilyina et al., 2013), and for the terrestrial biosphere on JSBACH (Giorgetta et al., 2013a). The data has been simulated with truncation of T31, which measures the horizontal resolution of the atmospheric model. This means the model has 31 levels (coarsest available resolution) and a low model top of 10 hPa (Stevens et al., 2013). For the ocean resolution a spatial resolution of three degrees has been used (GR30 model) (Stevens et al., 2013; Giorgetta et al., 2013b). The present-day conditions serve as initial conditions. Furthermore, the model is constraint by time-dependent topographic changes and river routing. The paleoclimate temperatures have been simulated for the last 26 ka, where 0 ka represent the present-day conditions and 26 ka the conditions 26 ka year before present-day.

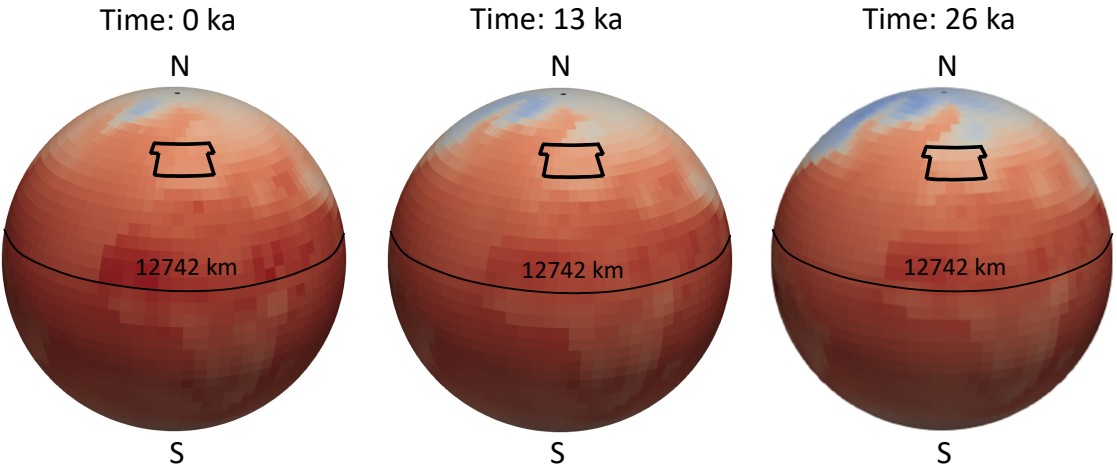

**Figure 1.** Paleotemperature data from the MPI-ESM Model for the timesteps 0 ka, 13 ka, and 26 ka. The black rectangle represents the outline of the surface temperatures used for the transient simulations of the CEBS model. Here 0 ka denotes the present-day conditions and 26ka the paleoclimate conditions from 26 ka before present-day.

## 3 Modelling results

### 3.1 The Central European Basin System (CEBS)

The study area is the Central European Basin System (CEBS) in northern and Central Europe. The CEBS is an intracontinental basin covering a domain that extends from the Tornquist Zone in the north-east and the Elbe Fault System in the south-west. This basin system underwent a multistage evolution of subsidence and uplift during its geological evolution since Permian times. Therefore, it results in a system of sub-basins (thickness up to 12 km) and a highly differentiated crust including a Precambrian crust of Baltica in the north-east, a Caledonian crust of Avalonia and Laurentia below the western and central

parts of the basin system, and the Variscan crust in the south (see Table A1 for more information on the different geological units comprising the 3D model). Several integrative studies (Maystrenko et al., 2013, and references therein) have imaged its present-day Permian to Cenozoic basin configuration as well as its deeper crustal and mantle structures to a high degree of confidence. Previous works by (Maystrenko et al., 2013; Scheck-Wenderoth and Maystrenko, 2013; Scheck-Wenderoth et al., 2014) have focused on integrating all available information into a gravity-constrained detailed 3D lithosphere-scale geolog-

ical model used as input for our modeling exercise. The model has a lateral extent of 1784x1060 km, and covers the whole sedimentary sequence from pre-Permian to Cenozoic, upper and lower, and the underlying mantle lithosphere down to the lithosphere-asthenosphere boundary (LAB). Table A1 lists the main geological features of interest as well as their respective rock properties, while Figures A1 and A2 illustrate the main geological units (as base maps and geological profiles, respectively).

### 3.2 Steady-State

In the following, we illustrate the influence of the thermal properties on the temperature distribution under a steady-state conductive thermal regime. In order to reduce the number of thermal properties that need to be investigated, we make a selection through various global sensitivity analyses for the steady-state conductive heat transfer. Therefore, the obtained results will

serve as a basis for the analyses that will be done in the following chapters. As the upper boundary conditions, we choose 8 °C, corresponding to the annual average surface temperature in the region.

First, we focus our investigation on the sedimentary layers. Hence, we only vary the thermal properties for the Cenozoic, Cretaceous, Jurassic, Triassic, Zechstein, Rotliegend, Permo-Carboniferous Volcanics, and Pre-Permian Rocks. All other thermal properties are kept constant, the values of which are derived from previous studies (listed in Table A1). Fig. 2 shows the re-

200 spective first- and total order indices. We observe that most contributions are first-order contributions and that the parameter correlations for the thermal conductivities are indeed negligible, that is, the differences between the first- and total order sensitivity indices (which define the parameter correlations) are below 5 %. Overall, we have a higher influence resulting from variations in thermal conductivity than from radiogenic heat production values. Therefore, we do not consider the correlation for radiogenic heat production since the absolute values are significantly below our threshold of 0.1. Note that the radiogenic

heat production is only included in future models to observe the changes due to transient effects. Therefore, we perform the

selection of the most prominent parameters based on the thermal conductivities with the given threshold of 0.1. Afterward, we retrieve the same number of radiogenic heat production values.

To narrow down the parameter space for further investigations, we make use of the five most influencing thermal conductivities and the five most influencing radiogenic heat production values (blue boxes in Fig. 2). We are interested in including the radiogenic heat production in our analysis despite its minor influence since we aim to investigate conceptual behavior changes induced by including paleoclimate information.

Hence, for the thermal conductivity, we consider the Cenozoic, Cretaceous, Jurassic, Triassic, and Pre-Permian Rocks. The thermal conductivity of the Cenozoic has the highest influence, followed by the thermal conductivity of the Cretaceous. The thermal conductivity of the Pre-Permian Rocks is slightly lower and the lowest sensitivity is found for the Jurassic and Triassic sediments.

In terms of heat production, the most influencing layer is the Pre-Permian Rocks sedimentary layer, followed by the Triassic sequence. The radiogenic heat production of the Cretaceous and the Cenozoic have a similar sensitivity, being significantly lower than for the sedimentary layers discussed above. The lowest sensitivity in terms of radiogenic heat production considered is associated with the Jurassic sedimentary layer.

The same analysis but considering thermal property variations in the crustal and mantle layers only is shown in Fig. 3. Here, we keep the thermal properties of the sedimentary layers as fixed. Overall, we observe analog to the previous analysis that the thermal conductivity has a higher influence than the radiogenic heat production. However, the difference in their influence is significantly lower. Again, the parameter correlations are mostly negligible, only the thermal conductivity of the Lower Crust shows higher differences of 14 % between first- and total order indices. For the crustal layers, we chose the three most influencing thermal conductivities and the three most influencing radiogenic heat production values for further analyses.

For the thermal conductivity, we consider the Upper Crust Baltica and Avalonia, and the Lithospheric Mantle. Here, the Upper Crust Baltica has the highest influence followed by the Upper Crust Avalonia. The Lithospheric Mantle has the lowest influence.

In the case of the radiogenic heat production, the highest influences are found within the Upper Crust Avalonia, followed by the Upper Crust Baltica. Furthermore, we consider the radiogenic heat production of the Lower Crust for further analyses. So far, our analysis has been based on sub-grouping the relevant units into either sedimentary layers of crustal-mantle domains. Therefore, we still did not investigate any possible parameter correlations among these units. We perform this analysis in the following, by systematically varying the most prominent thermal properties in both the sediments and the crust-mantle as derived from the previous analysis. We display the first- and total-order indices of this combined analysis in Fig. 4.

Based on the results from this first group of studies, we decide to focus on eight parameters in the following analyses. Our decision to narrow down the parameter space stems from the fact that both the construction of the surrogate model and the global sensitivity analysis are likely to become computationally too demanding if based on a large parameter space. The eight relevant parameters considered are (in descending order of relevance):

1. the thermal conductivity of the Lithospheric Mantle,

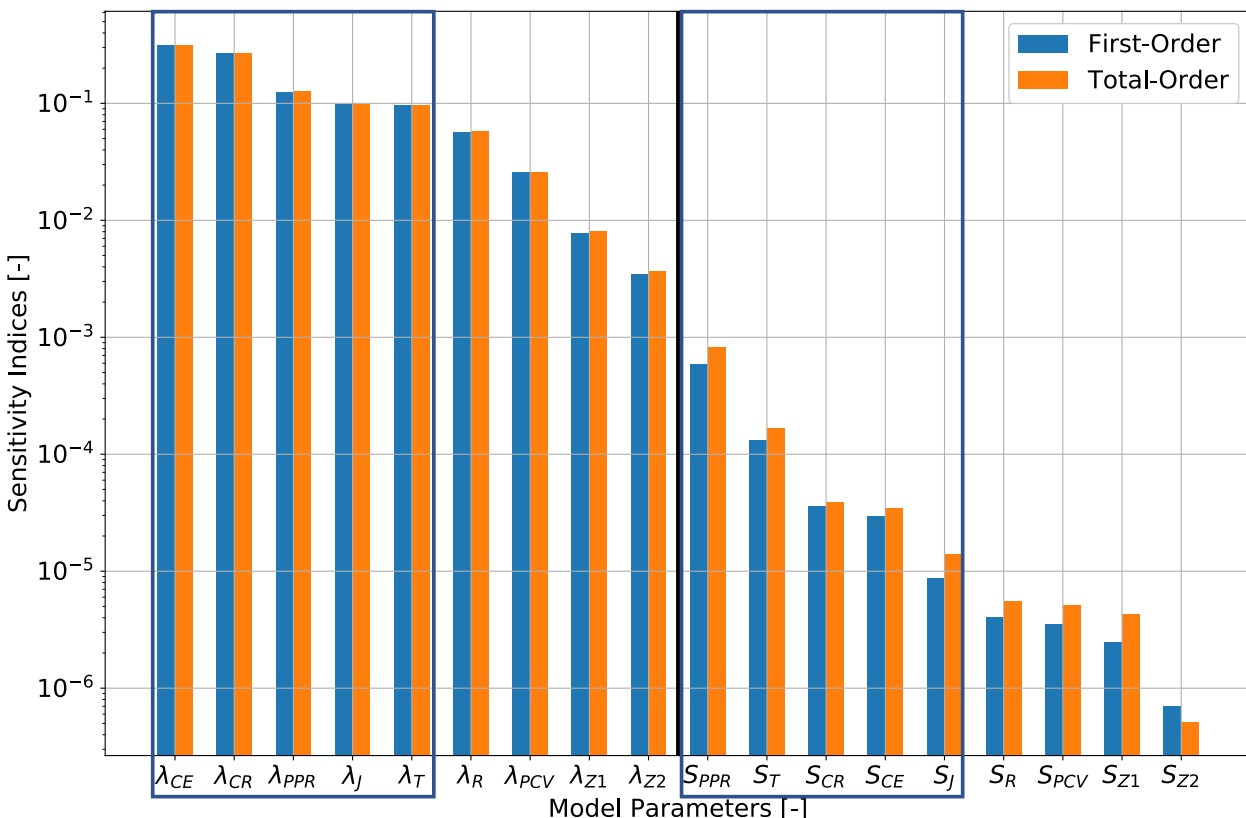

**Figure 2.** Global sensitivity indices for the analysis focussing on the sedimentary layers of the CEBS. The first-order indices are denoted in blue and the total-order indices in orange. Furthermore, we mark the parameters used in further analysis with blue rectangles and the vertical black line denotes the separation between the thermal conductivities and radiogenic heat productions. For the acronyms please refer to Tab. A1.

2. the thermal conductivity of the Triassic sedimentary unit,

3. the thermal conductivity of the Upper Crust Baltica,

4. the thermal conductivity of the Upper Crust Avalonia,

5. the thermal conductivity of the Cenozoic,

6. the thermal conductivity of the Cretaceous,

7. the radiogenic heat production of the Upper Crust Baltica and,

8. the radiogenic heat production of the Upper Crust Avalonia.

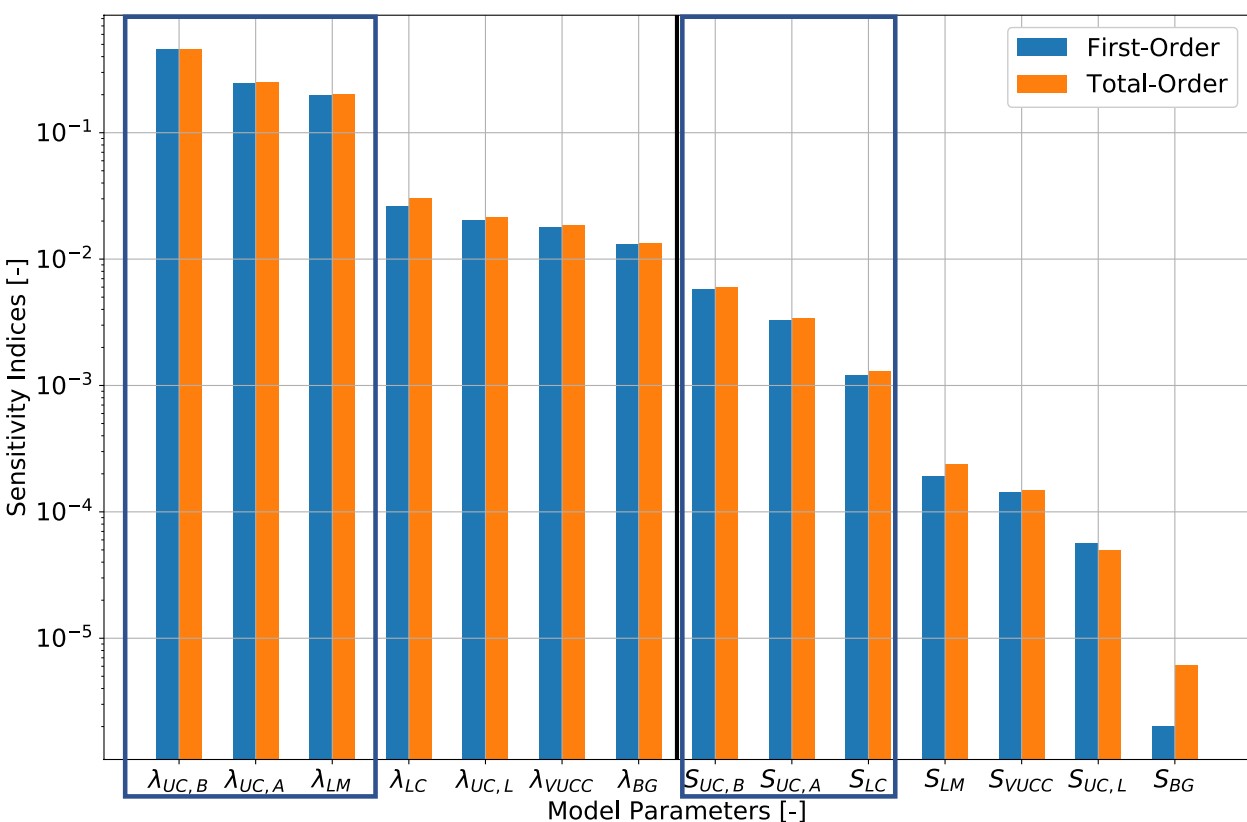

**Figure 3.** Global sensitivity indices for the analysis focused on the crustal layers of the Central European Basin System. The first-order indices are denoted in blue and the total-order indices in orange. Furthermore, we mark the parameters used in further analysis with blue rectangles and the vertical black line denotes the separation between the thermal conductivities and radiogenic heat productions. For the acronyms please refer to Tab. A1.

### 3.3 Impact of Solver Accuracy

In the previous section, we have quantified the impact of the various thermal properties for a steady-state run, considered as the "base" case for all further analyses. The investigations carried out so far have additionally enabled us to narrow down the parameter space on which to focus in the study to follow.

Before a detailed investigation of the influences of transient processes on the model response can be carried out, it is important to discuss the relevance of the accuracy chosen for the reduced model. For the steady-state simulations, all reduced models were constructed with an accuracy of $5 \cdot 10^{-4}$. This accuracy provides a global measure of the temperatures at every node of the model. Though local variations of accuracy can be considered, in this study we rely on a single global value. Our choice for this parameter ensures the reduced models to have a higher accuracy than that considered typical for temperature measurements (in the range of $10^{-2}$ to $10^{-3}$ °C). These same measurements have been used in previous works (Maystrenko et al., 2013; Scheck-

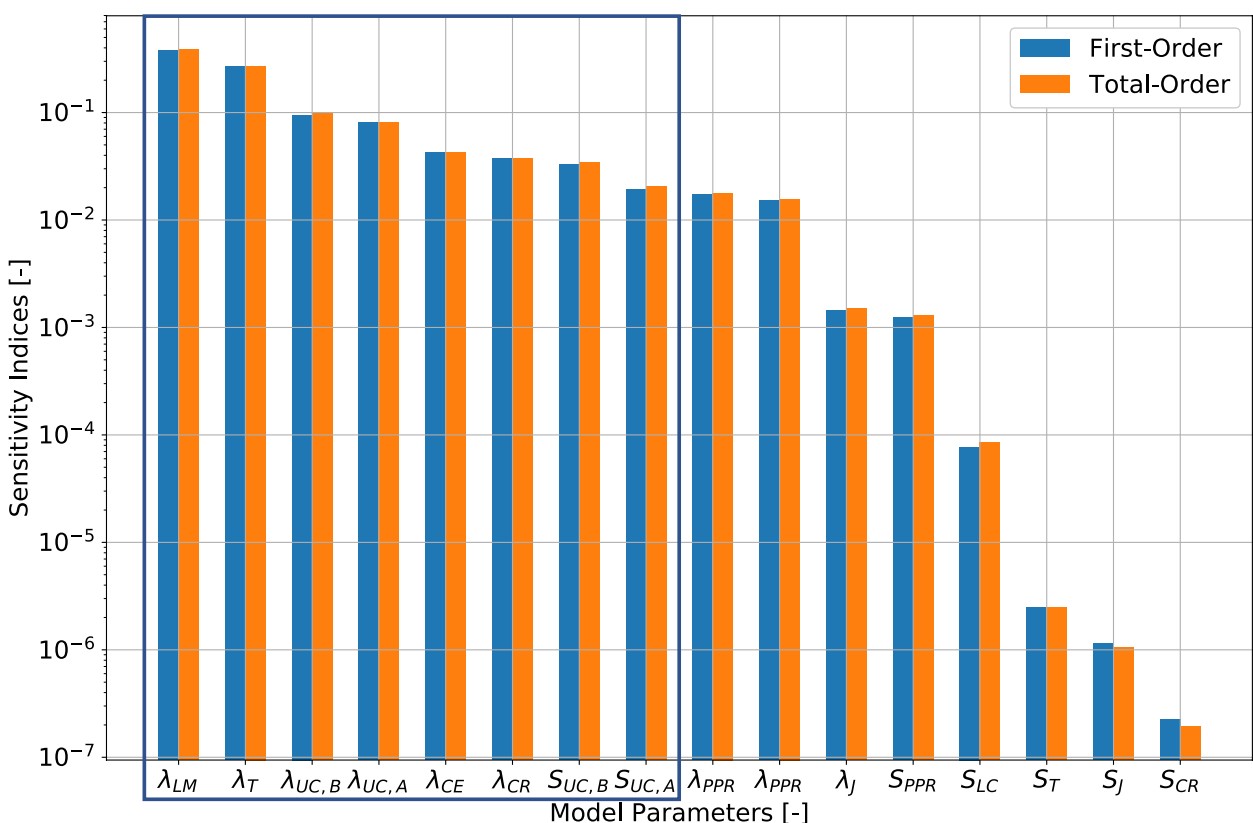

**Figure 4.** Global sensitivity indices for the analysis combining sedimentary and crustal layers of the Central European Basin System. The first-order indices are denoted in blue and the total-order indices in orange. For the acronyms please refer to Tab. A1.

Wenderoth and Maystrenko, 2013; Scheck-Wenderoth et al., 2014) to validate the model results, thereby we cannot infer any sensitive information below that accuracy. Therefore, for this specific case, the reduced and full models are equivalent to, for instance, parameter estimations and global sensitivity studies.

The RB method considers time as an additional parameter (Hesthaven et al., 2016), which increases the dimension of the parameter space when moving from a steady-state to a transient analysis. Besides, in a transient study, the system must be solved for each time step. Consequently, if we can assume a similar parameter complexity for the thermal model parameters, both the dimension of the reduced model and the compute time for each individual basis function will increase for the transient case. This, in turn, translates into a longer computing time for the sensitivity analyses.

We can compensate for this by relaxing the accuracy used in the reduced models ($4 \cdot 10^{-3}$) for the transient case. By utilizing such an accuracy, we are still able to obtain reduced models that have an error in the same order of magnitude as the temperature measurements , but with a significantly lower computational cost. Relying on a relaxed error tolerance could potentially introduce an additional error source. However, in our study, such a loss in accuracy should be considered insignificant. Indeed,

sensitivity analyses are based on the relative changes induced by model parameter variations. Since all simulations are affected

in the same manner by the chosen accuracy value, we can still maintain the same order of magnitude for such relative differences even if based on different accuracy levels (see Fig. 5).

To prove this point, we perform a sensitivity analysis focused on the sedimentary layers only by varying the adopted accuracy within the above-discussed bounds. The corresponding first- and total-order indices are plotted in Fig. 5. As Fig. 5 displays, for all thermal parameters that are considered for further analysis the results, despite the level of accuracy, are the same. Dif-

275 ferences are only limited to the parameters having the lowest sensitivity. However, these parameters must be excluded from the discussion since their errors of the sensitivity analyses are higher than their actual first- and total-order contributions. Also, the observed difference can likely be induced by the Sobol sensitivity analysis itself.

Based on what is stated above, we can conclude that, for the remaining analyses, we can make use of reduced models with a lower accuracy, thus having faster construction times of the reduced model and a less demanding sensitivity analyses.

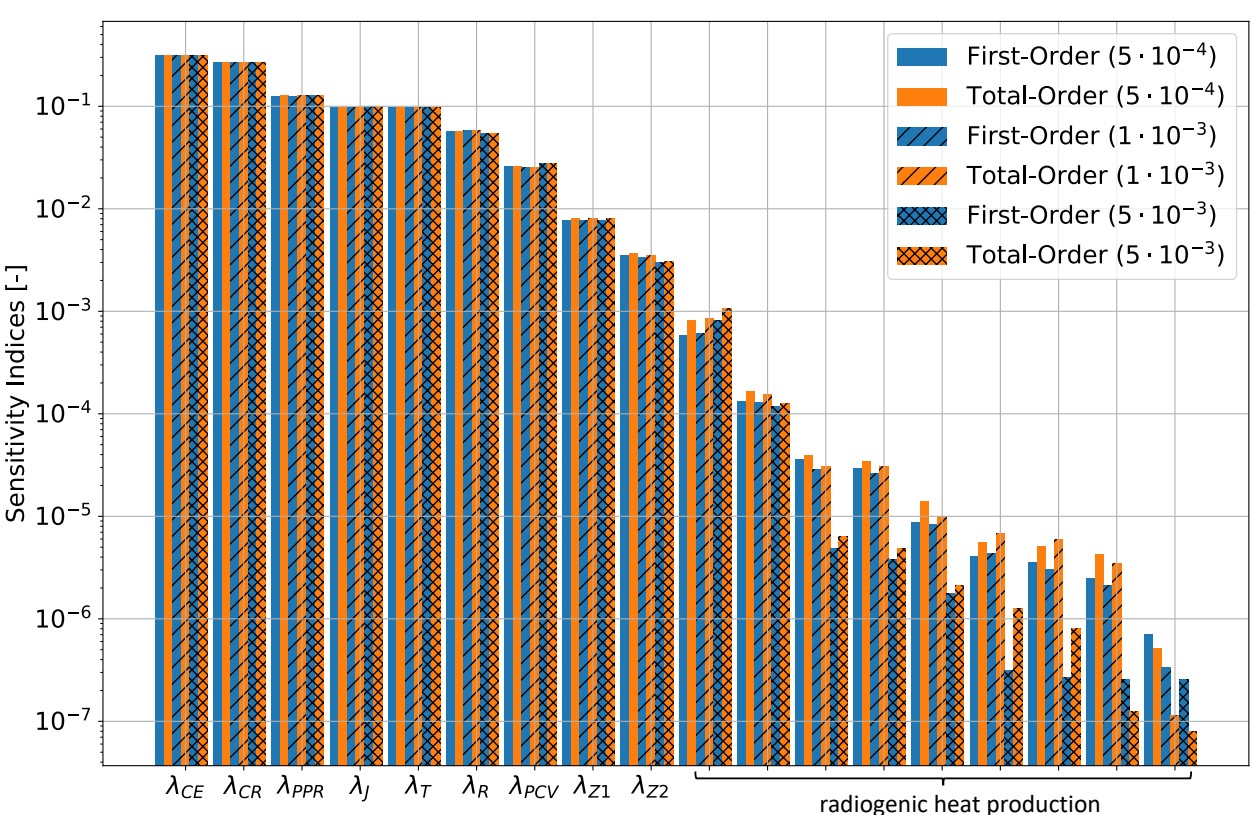

**Figure 5.** Global sensitivity indices for the analysis focused on the sedimentary layers of the Central European Basin System for an accuracy of the reduced model of $5 \cdot 10^{-4}$ (solid lines), $1 \cdot 10^{-3}$ (dashed lines), and $5 \cdot 10^{-3}$ (doted lines). The first-order indices are denoted in blue and the total-order indices in orange. For the acronyms please refer to Tab. A1.

 ## 3.4 Paleoclimate

In this study, we use several models for the transient case to investigate the influences of (i) considering time-varying surface boundary conditions as derived from paleoclimate models, and (ii) a change of the system dynamics from a steady to a transient state. Fig. 6 provides an overview of all transient models considered.

To include paleotemperature corrections to the steady-state results presented so far requires to consider a transient system.

Therefore, in a first step we perform a study to quantify the global influence derived from such a change in the system dynamics. For this reason, we perform a global sensitivity analysis based on a transient case, but without considering the effects of variation in the surface boundary conditions, i.e. without paleoclimate influence (Fig. 6 branch 1a).

After having quantified the influence of a change in the driving process itself on the sensitivity of the thermal properties, we can investigate the effect of incorporating paleotemperature information into our models (Fig. 6 branch 1b). In the following, we explain in detail how we account for paleoclimate corrections and which impact such corrections have on the respective sensitivities of the thermal rock properties.

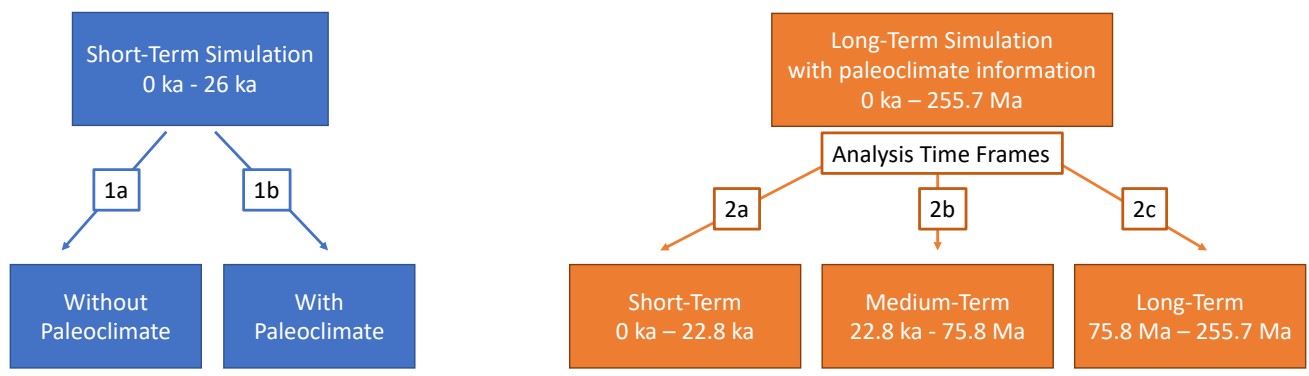

**Figure 6.** Schematic overview of the transient models.

### 3.4.1 Short-Term Transient Processes

For determining the impact of considering a transient simulation, we make use of a constant upper boundary condition, Dirichlet type of 1.6 °C (Fig. 6 branch 1a). This value corresponds to the average temperature derived from the paleotemperature values for our study area (Fig. 7). Such a value should be considered as an average over space and time. Therefore, we first take the average temperature values in space (orange curve in Fig. 7), and then we perform an additional average of this curve over time. The curve in Figure 7 displays variations in temperature between -8 °C and 8 °C. The resulting 1.6 °C from our temporal average results from the nonlinearity of the curve. We simulate the system under such thermal loading for 26,000 years which

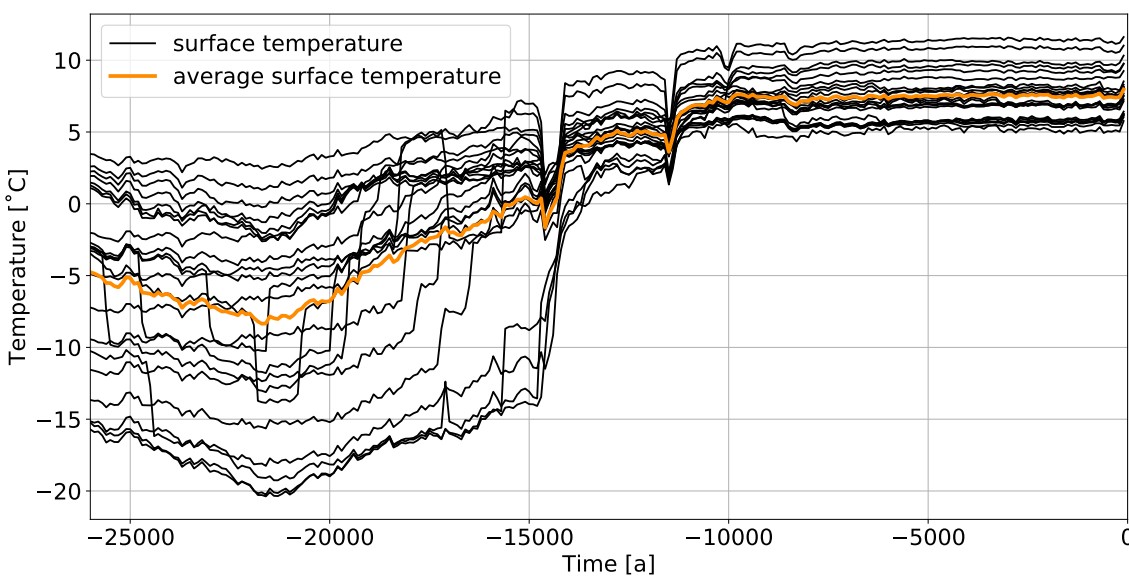

**Figure 7.** Paleotemperature reconstruction for all reconstruction points inside the CEBS model (denoted in black). Furthermore, we plot the average of all data points in orange.

is equal to the time frame for which reconstructed paleotemperatures are available. We adopted a constant time step size of
200 years. As the quantity of interest, we use, as for all following analyses, the total amount of heat in the model, that is, a cumulative value over all time steps.

Fig. 8 compares the sensitivities of the thermal properties for the steady-state and transient system with the respective initial and boundary conditions. Note that we plot the results of the transient analysis as a line plot despite the x-axis being discontinuous. We chose this representation to simplify a visual comparison. This said the continuous line has no physical meaning,
being only a visualization help to better capture the changes in the indices from parameter to parameter.

We observe that both the overall differences in the influence of the individual thermal properties and the parameters correlations increase. The Cenozoic and Cretaceous sedimentary layers gain significantly in importance, while the Triassic sediment maintains an influence similar in magnitudes as for the steady-state case. The Upper Crust is less significant in terms of both the diffusivity and the radiogenic heat production defined in Eq. 2. Note that we denote this radiogenic heat production divided
by the product of specific heat capacity and density in the following only as radiogenic heat production. The most extreme change is observed for the diffusivity of the Lithospheric Mantle which, for the steady-state runs, had the highest influence, but for the transient case one of the lowest. To sum up, we can observe a systematic change in the system response with the sedimentary layers gaining in importance, whereas the deeper crustal and mantle becoming less sensitive.

Regarding the correlations, we observe the strongest correlation between the diffusivities of the Cenozoic and Cretaceous

sediments, followed by the correlation between the diffusivities of Cretaceous and Triassic sediments. An aspect that is in agreement with the findings described above.

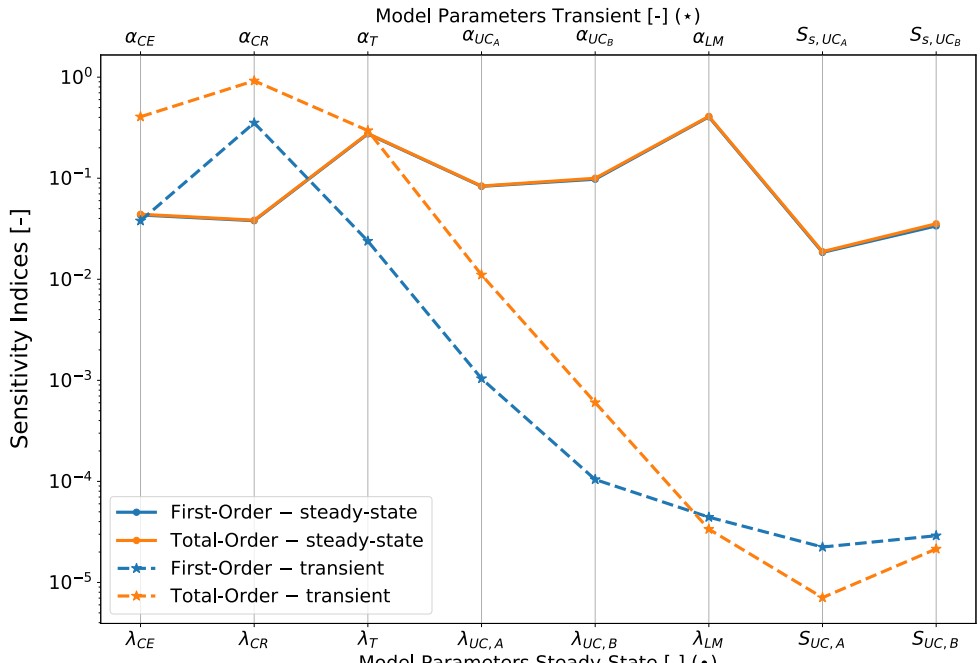

**Figure 8.** Global sensitivity indices for the analysis with steady-state (solid lines) and transient conditions (dashed lines) of the Central European Basin System. The first-order indices are denoted in blue and the total-order indices in orange. For the acronyms please refer to Tab. A1.

### 3.4.2  Data Fit

Prerequisite to investigate the influence of paleoclimate corrections on the thermal properties is a proper quantification of the sensitivity of the same thermal properties with respect to changes in the imposed upper boundary conditions. This is indeed
needed to rule out possible sources of uncertainty.

To properly quantify the influence on uncertainties in adopted paleotemperatures on the thermal properties of the different layers, we derive a first-order trend in surface temperature over time. Due to the current setup of the construction of the surrogate model this trend needs to be in the form of a smooth piece-wise linear function. Therefore, we cannot take the average temperature value as obtained from the raw data at each time step.
Uncertainties in the paleotemperatures arise from both temporal and spatial effects. Spatial uncertainties are mainly due to the low resolution of the paleoclimate data set when compared to the resolution of our input model. To derive the required trend,

we first focus on the past temperature distribution for all computational points of the global ESM analysis lying inside the CEBS area. Fig. 7 displays these spatial data points over time, black curves. Additionally, we also plot the average from all data points over time by the orange curve. By inspecting Fig. 7, it can be noticed how all considered points follow a similar trend. Therefore, we can consider the average of all data points as a good representation, and make use of it in the following to derive the paleotemperature trend to be imposed as a time-varying boundary condition. An overview of all tested fit for the upper boundary condition is provided in Fig. 9.

We fitted the paleotemperatures by a fourth-order polynomial (black line in Fig. 9), using the Python library SciPy (Jones et al., 2014). The final polynomial fit has the following form:

$$T_{\text{top}}(t) = 4.3 \cdot 10^{-8} t^4 - 2.7 \cdot 10^{-5} t^3 + 5.4 \cdot 10^{-3} t^2 - 2.8 \cdot 10^{-1} t - 3.6 \tag{3}$$

Note that the coefficients for the polynomials are normed to ka. We have tested additional polynomial degrees and used the L2 norm of the difference between the spatial average temperature (orange line of Fig. 7) to the fit to assess the quality. The third-order polynomial (blue line Fig. 9) cannot recover the paleotemperature pattern L2 norm of 18.1, and a fifth-order polynomial (green line Fig. 9) does not significantly improve the fit in comparison to the fourth-order polynomial (L2 norm decreases from 9.8 to 9.6). Furthermore, we tested to approximate the temperature with a sequence of polynomial fits (Fig. 9b) to better capture the minimum and maximum temperature values. This fit reduces the L2 norm to 6.0. However, due to the small influence of the paleotemperatures (as we will discuss in the following), we only use the fit with a fourth-order polynomial.

To investigate how sensitive the thermal properties are for uncertainties of the upper boundary condition, we use a time-variable scaling factor for the upper boundary condition. The lower and upper bound of the temperature variations arising from this scaling factor are displayed in Fig. 10. Additionally, the average is denoted by the orange curve and the actual data points over time are color-coded in light gray. We add a variation between $\pm$ 0.05 times the current time to the temperature fit as derived previously. We adapt the scaling factor, under the assumption that the uncertainties in the reconstructed temperatures should decrease while approaching present-day conditions. To derive the magnitude of the scaling parameter, we use the spatial distribution of the surface temperatures over time. In this way, we allow any physically plausible surface temperature variation. In a subsequent step, the trend has been normalized to the present-day surface temperature and applied to each point of the computational grid separately. This permits us to resolve the spatial distribution of the surface temperature. The paleotemperature data set comprises the last 26 ka. As such, it also contains data from the latest Weichsel-Glaciation. The displayed temperature curves represent, during glaciation times, the temperature at the top of the ice sheet. Hence, we would normally need to correct the temperatures to obtain the values at the bottom of the ice sheet. Note that by applying a fourth-order polynomial fit, we already implicitly account for this. By applying, in addition, a scaling factor as a function of time, we can also account for all possibly remaining correction terms.

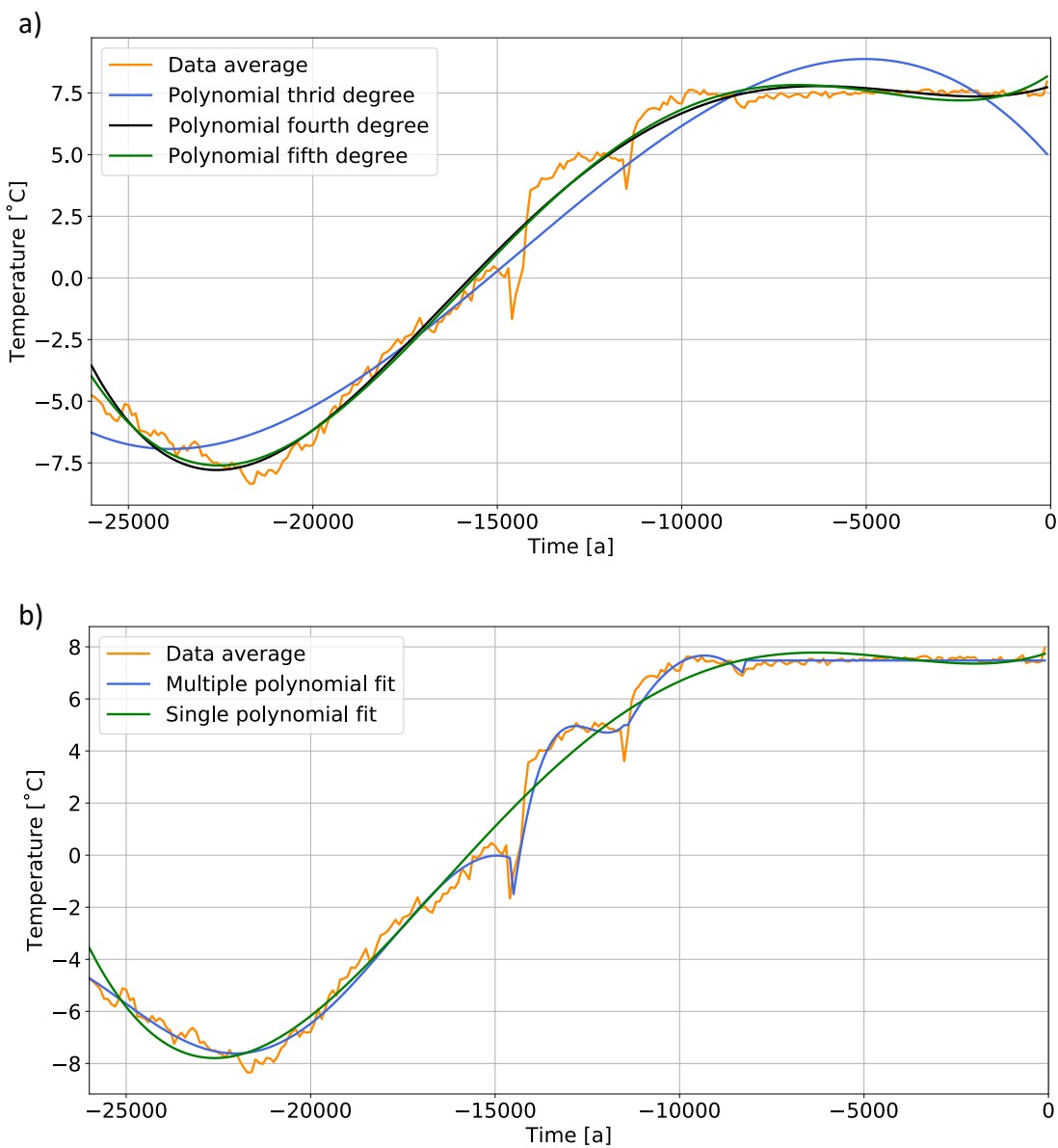

**Figure 9.** Trend for the paleotemperature upper boundary condition of the CEBS model using a) single polynomial fits and b) single and multiple polynomial fits.

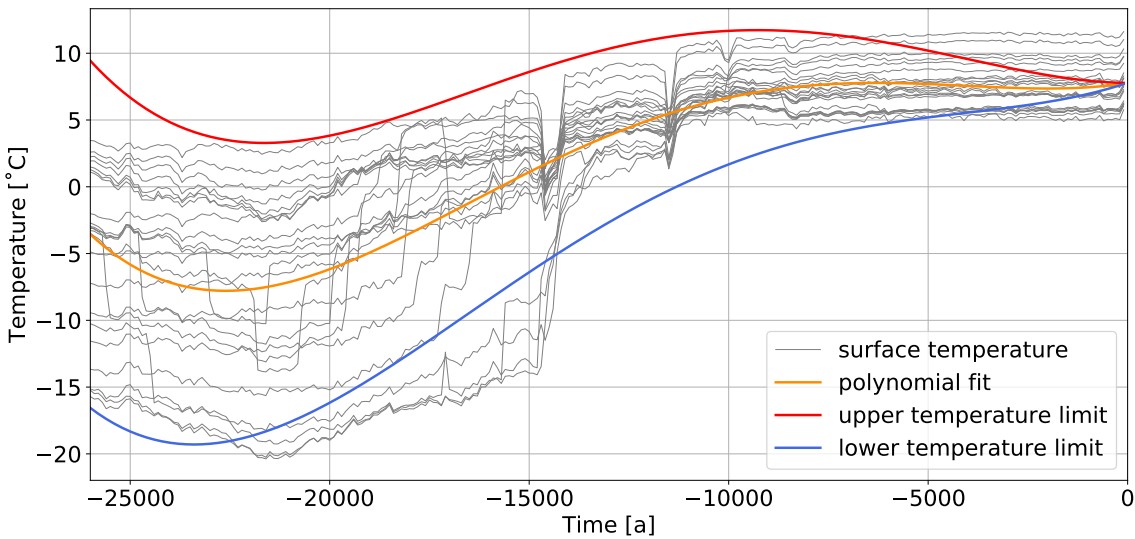

**Figure 10.** Trend for the paleotemperature upper boundary condition of the CEBS model.

### 3.4.3 Transient Boundary Condition

In Fig. 11 we compare the transient simulation with a constant upper boundary condition (solid lines – Fig. 6 branch 1a) and with a time-dependent (paleoclimate correction) boundary condition (dashed lines – Fig. 6 branch 1b). We observe for the first four thermal properties (diffusivity of the Cenozoic, Cretaceous, Triassic, and Upper Crust Avalonia) no significant changes for either the first- or total-order contributions between the two models. All remaining parameters have insignificant first- and total-order contributions. Note that we cannot discuss the difference between the two models for these remaining parameters because the confidence intervals of the sensitivity indices are larger than the indices themselves. Furthermore, it is interesting to note that the scale factor, which we used as a measure of the uncertainty of the upper boundary condition, has one of the lowest sensitivities. Although, the transient boundary conditions do not play a significant role in the current setup, transient physical processes do influence the model parameters significantly, as we will discuss in the next section. In this regard, we should note that such a low influence of the boundary condition mainly arises from the fact that we use first-order Dirichlet boundary conditions. This will be further discussed in the Discussion Section.

### 3.5 Transient Processes

So far, we have limited our analysis to short-term transient processes (Fig. 6 branch 1) and their sensitivities of the thermal properties. In this section, we focus on the long-term period (Fig. 6 branch 2). Therefore, we increase the simulation time from 26 ka to 255.7 Ma. To maintain affordable computing time, we adopted a different time discretization, where the initial

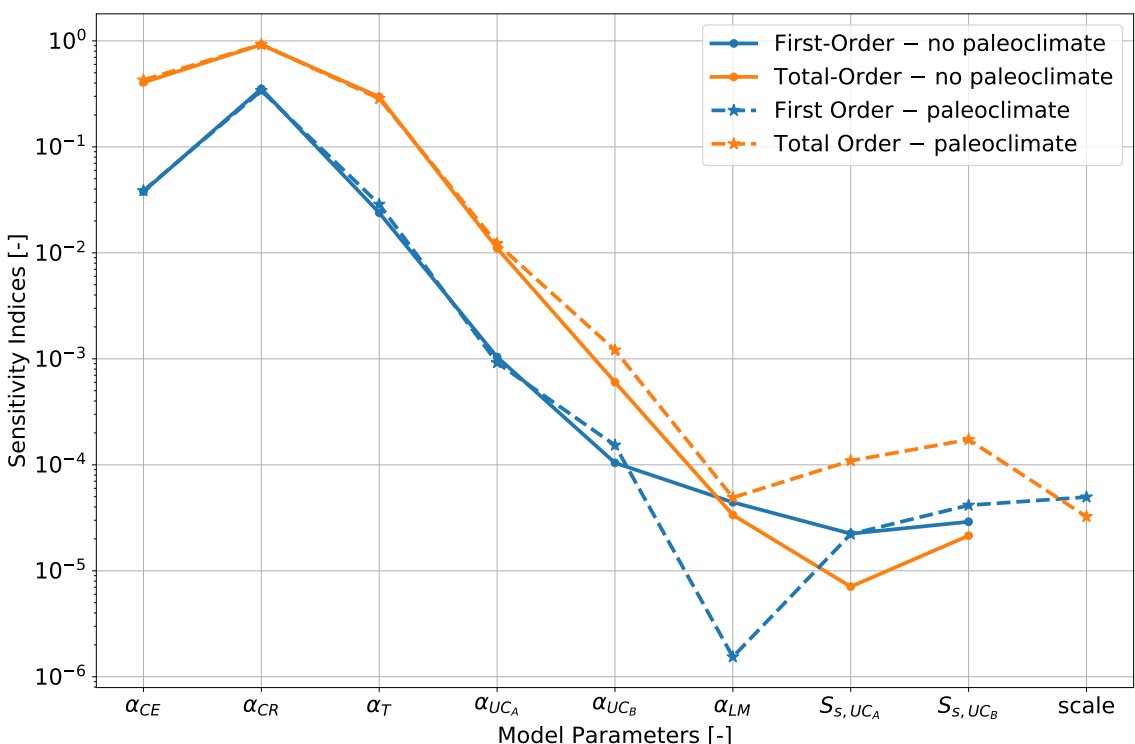

**Figure 11.** Global sensitivity indices for the analysis including paleotemperature information for the upper boundary condition. The first-order indices are denoted in blue and the total-order indices in orange. For the acronyms please refer to Tab. A1.

time step of 2 ka increases linearly by a factor of 1.5 upon each successful transient run. We perform four sensitivity analyses
considering: the entire time period, and the periods between:

- – 0 ka and 22.8 ka (Fig. 6 branch 2a)

- – 22.8 ka and 75.8 Ma (Fig. 6 branch 2b)

- – 75.8 Ma and 255.7 Ma (Fig. 6 branch 2c)

This subdivision has been chosen to differentiate between short-term, mid-term, and long-term physical effects. In Fig. 12, we
display at the top the results considering the entire time period. Along the bottom of the same figure, we portray the results of
the different analyses done, that is, short-, mid-, and long-term period, from left to right, respectively.
First, we discuss the results considering the whole simulation time. A second discussion point is on how the sensitivities evolve.

By comparing the total-order contributions among all analyses, we can observe that the diffusivities of the Cenozoic, Cretaceous, Triassic, and Upper Crust Avalonia have a similar contribution despite the time window adopted. For the remaining
parameters, the long-term total-order contributions are between the short-term and steady-state total-order sensitivities. Except for the diffusivity of the Upper Crust Baltica, all thermal parameters have long-term contributions close to their short-term, but they do differ with respect to the steady-state contributions.

In terms of correlations, we observe a slight decrease in the correlations concerning the short-term analysis. Still, the correlations are significantly higher than those observed for the steady-state scenario. The highest correlations occur between the
diffusivities of the Cenozoic and Cretaceous sediments and between the diffusivities of Cretaceous and Triassic units.

Overall, the main influences can be noticed for the diffusivities of the sedimentary layers. However, the diffusivities of the crustal layers are significantly increased with respect to the short-term analysis. The influence of the diffusivity of the Lithospheric Mantle and the radiogenic heat production remains negligible in all cases. In addition to the consideration of the entire

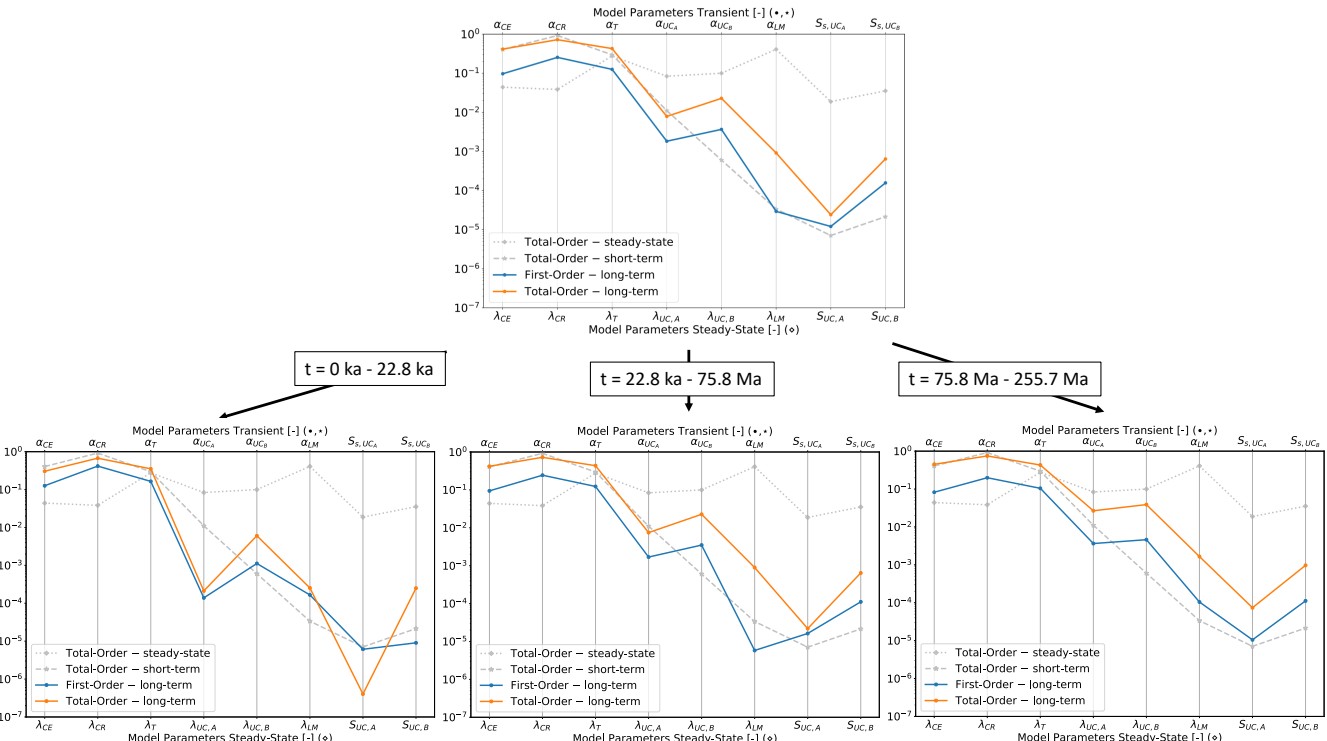

**Figure 12.** Global sensitivity indices for the analysis considering a simulation time of 255.7 Ma. The first-order indices are denoted in blue and the total-order indices in orange. Additionally, the total-order indices of the short-term (dashed gray line) and steady-state analyses (doted gray line) are plotted. For the acronyms please refer to Tab. A1.

simulation time frame, it is interesting to investigate how the sensitivities evolve throughout the simulation within specified
time windows. Therefore, we subdivided the whole time frame into three different windows. The first period (Fig. 6 branch 2a),

from 0 ka to 22.8 ka corresponds to the short-term analysis presented in Section 3.4.1. Here, we observe major differences for the changes of the crustal diffusivities. The diffusivity of the Upper Crust Avalonia has lower indices, whereas the Upper Crust Baltica has higher sensitivity indices. Furthermore, we can notice a drop in all correlations. Again the highest correlations occur between the diffusivities of the Cenozoic and Cretaceous sediments, and between the diffusivities of Cretaceous and

Triassic.

We observe a significant change from the first to the second period (22.8 to 75.8 Ma, Fig. 6 branch 2b). The influence of the crustal diffusivities and the Lithospheric Mantle significantly increases, this is especially the case for the Upper Crust Avalonia. Also, the crustal radiogenic heat production gains in importance. In contrast, no changes can be noticed in the sensitivities indices of the sedimentary diffusivities. Overall the sensitivity indices still follow the trend of the short-term analysis. At the

same time, the parameter correlations are also similar to those obtained for the first period. For the second period considered, we can conclude that while the crustal layers gain in importance, the sedimentary layers do not show any systematic variations. Moving to the third and last period (75.8 Ma to 255.7 Ma, Fig. 6 branch 2c), we again observe some significant changes. The thermal properties of the Upper Crust become more important as well as the diffusivity of the Lithospheric Mantle, which has now higher sensitivity indices. In contrast, the crustal diffusivities resemble those of the steady-state analysis. The sensitivity

indices of the sedimentary layers remain unchanged. The highest parameter correlation can be found among the diffusivities of the Cenozoic and Cretaceous sediments.

## 4   Discussion

In the following, we open a discussion on the results obtained for the steady-state model and those found while considering

a transient system. Note that all results of the sensitivities analyses presented are highly dependent on the quantity of interest chosen. Therefore, we must call for caution while discussing these results and use comparable quantity of interests throughout the entire paper.

### 4.1   Steady-State

The sensitivities of the steady-state model are mainly controlled by a combination of the volumetric contributions of the indi-

vidual layers' thermal properties. Generally, the thermal conductivity has a significantly higher influence on the total amount of heat than the radiogenic heat production, which does follow our expectations. The only layers for which the radiogenic heat production has some significant influence are the Upper Crust layers. For the sediments, the differences between the relevance of the role of the thermal conductivity and the radiogenic heat production are higher than for the crustal layers. This is caused by the higher radiogenic heat production of the latter rocks.

Lower thermal conductivities yield a higher impact on the model response since a lower thermal conductivity results in a larger amount of heat stored within a layer, i.e. blanketing thermal effect. This is also the reason why the Zechstein layers have a smaller impact although being relatively thicker than the other sedimentary units. In contrast, the Lithospheric Mantle has a

relatively prominent influence although it has a high thermal conductivity. This is because this layer counts for most of the total system volume (around 76%, see Table A1).

Higher heat production values yield a higher influence since they result in a larger amount of generated heat. That is the reason why the influence of the Upper Crust is so significant.

The steady-state analyses have only negligible higher-order contributions, as apparent by the not statistically significant difference between the first- and total-order contributions. This means that we have no significant parameter correlations.

## 4.2 Paleoclimate

In the following, we use the steady-state sensitivities as our base scenario and discuss the results of the transient case only in terms of relative changes with respect to the latter scenario.

### 4.2.1 Short-Term Transient Processes

Considering a simulation time of 26,000 years (short-term window, Fig. 6 branch 1a), we observe that the model response is most sensitive to the diffusivity of the sedimentary layers. Both the diffusivity of the crustal and mantle layers and their
radiogenic heat production do not influence the model response. This can be related to having considered a finite temporal extent. Given the thermal properties typical of crustal and mantle rocks, and their respective thickness, thermal equilibrium cannot be reached within the allotted time of 26,000 years. To better illustrate that we show in Fig. 13 the difference in the temperature distribution of the simulation after 26 ka and the initial condition for the upper 30 km. Heat transfer occurs mainly in the uppermost layers (closer to the surface boundary conditions), hence their great impact. As demonstrated by the steady-
state analyses, the radiogenic heat production in these layers does not have a significant influence. Given the short time window considered, imposed variations in the surface boundary conditions could not diffusive into the crust. Therefore, the model is relatively insensitive to any variations in either the diffusivities or radiogenic heat production for those deeper layers.

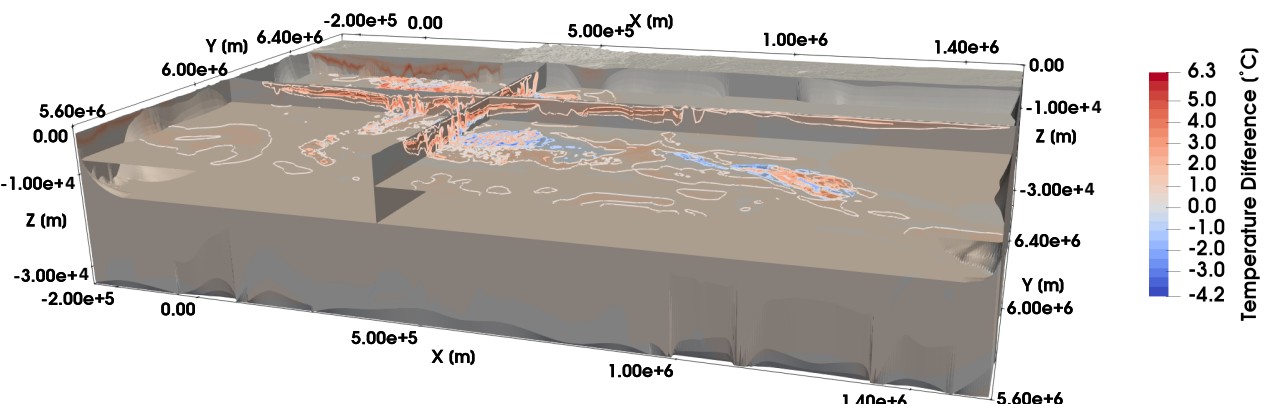

**Figure 13.** Difference in the temperature distribution for the upper 30 km between the state at time 22.8 ka and the initial condition for the transient simulation considering a simulation time of 255.7 Ma.

### 4.2.2 Transient Boundary Conditions

We observe no differences in the sensitivities of the thermal model parameters when considering paleoclimate corrections by means of the adopted transient boundary condition. Furthermore, the model is insensitive to the scaling factor for the upper boundary condition, which accounts for possible uncertainties of the boundary condition. Note that this scaling factor allows variations of up to 15 °C. Hence, we already allow every physically plausible temperature along the upper boundary condition. Additionally, we account for increasing uncertainties over time (from present-day backward).

The reason why the model is insensitive to changes in the upper boundary condition is likely related to the chosen setting. For both the time-dependent and -independent case, we apply a Dirichlet type constraint. A Dirichlet constrain forces the model to have a defined and prescribed value of the unknown variable at the respective boundary. Additionally, we also account for the basal boundary conditions in terms of a prescribed temperature. Consequently, with this set of constraints, we fix the total amount of heat in the model that corresponds to our quantity of interest. Although we carried out the investigations based on relative changes between the different simulations, we observe no variations. This is because we pre-define, through the type of boundary conditions, the total amount of heat in the system.

At the current stage, this is unavoidable since we can neither define a different meaningful quantity of interest nor a different type of boundary condition to be imposed as derived from global ESM models. Classically, the quantity of interest is defined as the norm of the misfit between simulated and measured temperature values. However, this is in our case not possible since we are interested in changes over the whole model. This is especially relevant while investigating the sensitivity of parameters for rocks buried at greater depths (higher than a few kilometers) where we lack any temperature measurements against which to compare the obtained results. This would then lead to a bias in the final estimates, with deeper regions being systematically under-represented in terms of their plate-like influence. Additionally, it is worth mentioning that we can only rely on direct measurements for the present time. Hence, the influence of past time steps would be also underestimated accordingly.

A possible solution to this would be to adopt in future studies a different upper boundary condition, moving from a Dirichlet to a more representative Robin-type constraint. This would improve the global sensitivity analysis also in relation to the physics occurring at the surface interface since it would enable us to consider proper interactions between the atmosphere and the earth's subsurface. The use of a Robin boundary condition, however, would require to have detailed information about the heat in- and out-flux across this interface. This is currently not possible and would require a dedicated effort and closer interactions between the climate and subsurface communities.

### 4.3 Transient Processes

Although both "short-term" analyses (Fig. 6 branches 1a and 2a) consider a comparable time frame their sensitivities differ significantly for the influences arising from both the diffusivity and radiogenic heat product of the Upper Crust Avalonia. Note that in this paper, we use the expression "short-term" for all simulations with a time frame smaller than 26,000 years.

To investigate the reason for this difference, we compare the time-stepping approaches of both simulations. The short-term analysis conducted for the investigation of paleoclimate effects (Fig. 6 branch 1b) uses a constant time step size of 200 a,

resulting in 130 time steps for a simulation time of 26 ka. In contrast, the other short-term investigation (Fig. 6 branch 2a) uses an initial time step size of 2 ka that grows linearly by a factor of 1.5. Due to the different time-stepping approaches, the paleoclimate short-term analysis has equally distributed "snapshots" over time. The short-term analysis that makes use of a non-constant time stepping provides more snapshots at later periods. Therefore, the thermal properties that become important later on in the system appear more pronounced than in the paleoclimate short-term analysis.

The logical consequence would be to use only constant time step sizes for the sensitivity analyses to avoid any bias by the time-stepping method. However, this is for long simulation periods unfeasible since it would result in unaffordable computational costs. Another possibility is to introduce a weighting scheme to compensate for the bias introduced by the time stepping. Since this paper aims to investigate the influences of transient processes, in general, this is not of primary concern here. Nonetheless, it would be interesting to investigate this phenomenon in future studies.

Focusing on how the sensitivities change over time, we observe that for the short-term period (Fig. 6 branch 2a) mainly the diffusivities of the sedimentary layers have an impact on the model response. For the second period (Fig. 6 branch 2b), the influence of the crustal and mantle diffusivities gain in importance, whereas the impact of varying radiogenic heat productions remain negligible. We consider a conductive heat transfer problem with the radiogenic heat production as the source term. Hence, we take a diffusive dominated process into account. Additionally, we have a cold upper and a warm lower boundary condition, resulting in a temperature gradient that increases with depth. Hence, the heat in the system is transported from the lower to the upper boundary. Therefore, the sedimentary layers, which are located at the uppermost part of the CEBS has a relatively prominent influence on the short-term system dynamics.

At longer time, the "heat-signal" is transported over longer distances within the plate. Again, for a better illustration, we display in Fig. 14 the difference between the simulations at 10 Ma and 583.9 ka. Consequently, the crustal and mantle diffusivities grow in relevance. These observations also imply that we can use the sensitivity analyses to investigate specific regions of interest where heat transfer is active and, therefore, how any thermal signal (perturbation to a steady background state) propagates over time.

The radiogenic heat production has the highest impact in the last period analyzed. This is indicative that, during the whole evolution considered, the system could equilibrate by diffusion This aspect is schematically illustrated in Fig. 15 where we show the computed differences in the temperature distribution at 75.8 Ma (beginning of long-term period) and 255.7 Ma (end of long-term period). Even for the last period (Fig. 6 branch 2c), the sensitivities of the diffusivity of the Lithospheric Mantle and the crustal radiogenic heat productions are not as high as for the steady-state scenario. This can be partially associated with the fact that we have not yet reached a full equilibrium, that is that we are looking at a dynamic equilibrium in the system evolution. There is an additional reason for this discrepancy observed. For the transient sensitivity analysis, we can face two options: either we consider the entire simulation period or only a portion of it. If we take into account the entire time frame, we never get a sensitivity distribution close to the steady-state scenario since we incorporate early and late time steps and hence get a weighted average over the sensitivities, where the weighting depends on the number of time steps. If, in turn, we consider only a certain time window, i.e. the very last time steps, we will still not get a representation of the steady-state system. This is because a steady-state system does not consider any time-dependency in its formulation. This is in contrast to a transient

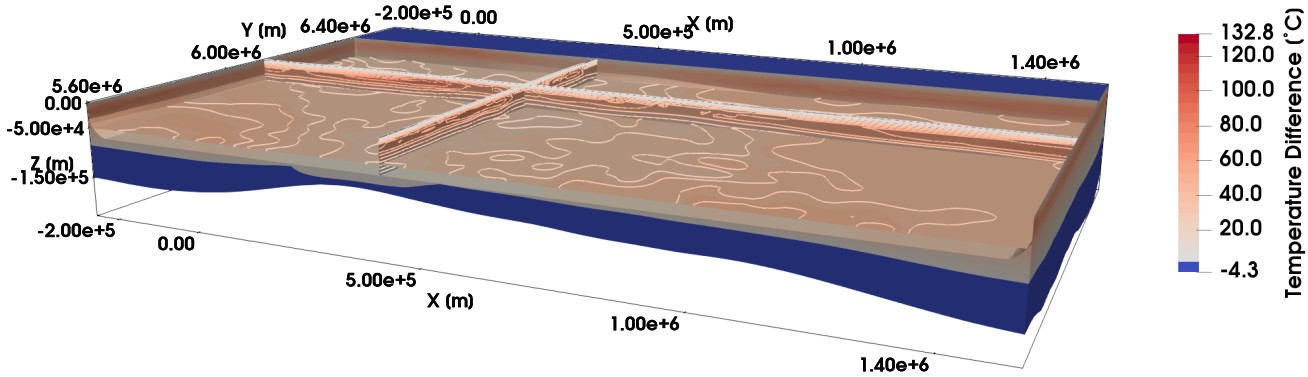

**Figure 14.** Difference in the temperature distribution between the state at time 10.0 Ma and time 583.9 ka for the transient simulation considering a simulation time of 255.7 Ma.

system, where these effects even if small are taken into account. If we compare the temperature distributions at this final period, we would observe no significant differences. However, the sensitivity analysis investigates the relative changes within a certain system. Therefore, also small changes in magnitudes can lead to a higher than expected sensitivity. Therefore, our analysis also bought us to conclude that in any geological system, we will never reach a final thermal equilibrium, but rather we will
always observe variations from such an equilibrium, even if small in magnitudes. Only the consideration of these variations could enable us to obtain equal sensitivity distributions for steady-state and transient simulations.

The analysis considering the entire simulation time is almost identical to the analysis for the second period.

The subdivision into three time periods has the aim to identify short, medium, and long-term transient effects. Since the diffusion of a thermal signal requires several million years to propagate throughout a typical lithospheric plate, the effects of
any spatial distribution of internal heat sources can only affect the temperature configuration at a later stage in the evolution of the system. This also enables us to subdivide the whole time evolution into different stages following the dominant physics.

The early period (short-term analysis above) is chosen such that it matches the time frame from the paleoclimate analysis. The third period contains the very last time steps, where visual changes due to the radiogenic heat production occurs. The second period contains all remaining time steps. Consequently, the different periods comprise a different number of time steps.
The first consists of eight, the second of 19, and the third of four time steps. Since the second period has significantly more time steps than the other two it majorly influences the entire sensitivity distribution (considering the whole simulation time). Furthermore, the parameter correlations are significantly higher for the transient than for the steady-state simulations. In the transient case, the highest correlations always occur between the diffusivities of the top geological layers, which is caused by their overall high impact on the model response. The parameter correlations are higher for the transient case since for this latter
case we have a propagation of a colder temperature front towards the bottom of the model due to the imposed upper Dirichlet boundary condition. Therefore, the interaction between the layers becomes more important.

Note that we used a stationary geological model throughout the entire paper. Thus, we do not consider the geological evolution

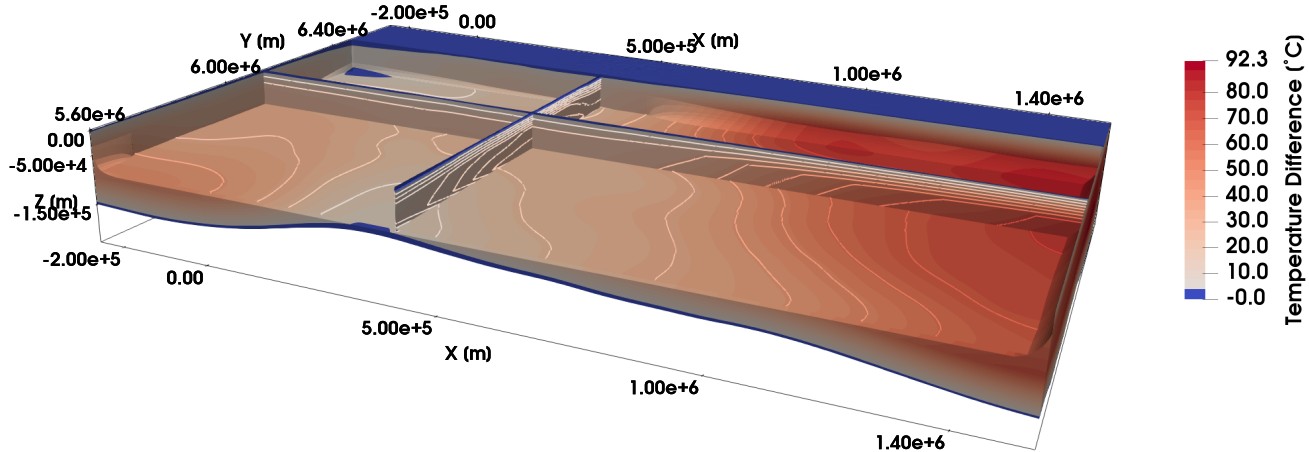

**Figure 15.** Difference in the temperature distribution between the state at time 255.7 Ma and time 75.8 Ma for the transient simulation considering a simulation time of 255.7 Ma.

of the model over time. The reason for this is that we want to highlight the changes in the sensitivities induced by transient effects. Considering at the same time changes of the geological model would mask these effects. Indeed, to allow variations
in the input geological model over time would have increased the parameter space in a rather unpredictable manner, thereby resulting in an over-parameterized problem. In addition, we would be asked to account for uncertainties in the backward reconstruction of the model, which is based on available present-day geological information. All these aspects would have hindered a proper assessment of the model outcomes and of the consequences of the physical processes that we targeted in the study. As pointed out throughout the paper, this study is methodological. We use the CEBS model as our case study but the
intention is to provide an ideal setup for the investigation of the stationary versus a non-stationary case rather than providing an ideal geological description.

## 5   Conclusions

We presented in this paper a quantitative framework for determining the impact of transient processes, including paleoclimate boundary conditions, on conductive heat transfer problems for basin-scale models.
Transient processes have a significant influence on the sensitivity distribution, generally leading to higher impacts of the sedimentary layers. Furthermore, the sensitivities are influenced by the time-stepping approach and by the simulation time. Hence, it is important to consider this in the analysis and carefully chose an appropriate time-stepping method and simulation time to avoid biased results. It is furthermore advisable to perform separate analyses for portions of the simulation time to investigate how the sensitivities of the thermal properties evolve.
Furthermore, we have demonstrated how a global sensitivity analysis can be used as a tool for improving our understanding of the subsurface. Relying on a chosen quantity of interest, we are able to design a sensitivity analysis that focuses on the physical

processes instead of the measurements (as commonly done). With this setup, it is possible to determine the most influential physical processes acting at a specific time in the system evolution and its depth and spatial extent.

Next to illustrating the influence of the transient process itself, we also investigated the influence derived from considering time-varying Dirichlet boundary conditions. We have been able to demonstrate that such a boundary condition set up is not advisable to properly incorporate paleotemperature information. We have been able to show that the thermal properties (their sensitivities and correlations) are not impacted by the changes in the upper boundary condition. However, this aspect should not drive to the conclusions not to consider any interaction between the atmosphere and the earth's subsurface. On the contrary, for conductive heat transfer problems, this would call for additional efforts in better integrating these dynamics by a more physically consistent set of boundary conditions (Robin boundary condition) which would describe the natural process occurring at this interface in a more quantitative and reliably manner. The lack of our knowledge about heat in- and out-flux and their variations in space and time asks for ongoing research efforts.

In this paper, we used global sensitivity analyses instead of local analyses to not only investigate the influence from the thermal parameters themselves but also their correlations (Sobol, 2001; Wainwright et al., 2014; Degen et al., 2020a). Another disadvantage of local sensitivity analyses is that they tend to overestimate the impact of the individual model parameters, as shown in previous studies (Degen et al., 2020a). Global sensitivity analyses have the disadvantage of being computationally expensive. Thus, we show that we overcome this issue through the construction of a surrogate model. Since we are interested in the entire temperature distribution and not only in the temperatures at pre-defined measurement locations we use the reduced basis method to construct our surrogate model. In contrast to other surrogate models, it is not restricted to the observation space (Miao et al., 2019; Mo et al., 2019).

To showcase the need for a surrogate model, we should detail the total number of forward simulations required in our study. In total, we performed 12,000,000 steady-state forward simulations with an average compute time varying between 1 ms and 4.2 ms for a single forward simulation of the various model scenarios. For the transient analyses, we require 560,000 forward simulations with an average duration of 13 ms to 200 ms for the various model realizations. If we would have used the finite element method instead, these investigations would have been infeasible since even on a high-performance infrastructure we would obtain simulation times in the order of minutes for the steady-state simulations and hours in case of the transient ones.

To conclude, the combination of global sensitivity analysis and surrogate modeling have helped us to demonstrate the relevance of considering transient aspects in subsurface thermal studies, an aspect that has been too often neglected so far. Transient processes yield significantly differing influences of the thermal properties than steady-state simulations. To achieve this, it is advisable, if not mandatory to simplify the problem on its mathematical level rather than on the physics at play, as we demonstrated in our study.

*Code availability.* For the construction of the reduced models, we used the software package DwarfElephant (Degen et al., 2020b, c). The software, which is based on the finite element solver MOOSE (Alger et al., 2019), is freely available on Zenodo (https://zenodo.org/badge/latestdoi/117989215). The sensitivity analyses are performed with the Python library SALib (Herman and Usher, 2017).

.

*Data availability.*   The global paleoclimate data have been generated with the MPI-ESM model code, which is freely available to the scientific community and can be accessed with a license on the MPI-M model distribution website (http://www.mpimet.mpg.de/en/science/models). The global paleoclimate data set is owned by the Max Planck Institute and can be obtained on request (publications@mpimet.mpg.de). The derived temperature trend used in this study (Section 3.4.2) is freely available together with all information to understand and reproduce

the results from the paper. The 3D geological model of the CEBS, is available from https://doi.org/10.1016/j.tecto.2013.04.023. The link to the archive is automatically regenerated upon clicking on "Download all supplementary files". The data for the refined sedimentary sequence with respect to the above described model as adopted in this study is available as DOI and online material via the following link https://doi.org/10.5880/GFZ.4.5.2020.006.

**Appendix A: Geoological and Rock Properties of the CEBS model**

**Table A1.** Geometrical and rock phsical properties of the different units integrated in the 3D structural model. Symbols listed: h (av)=average unit thickness, h (max)=maximum unit thickness, λ=thermal conductivity, $H$=heat production rate, $\rho$=density, and $c_p$=rock heat capacity. The unit volume has been computed based on the average thickness of each unit.

| Acronym | Layer | main lithology | h (av) | h (max) | volume | $\lambda$ | $H$ | $\rho \times c_p$ |
|---|---|---|---|---|---|---|---|---|
| - | - | - | $km$ | $km$ | $10^5 km^3$ | $Wm^{-1}K^{-1}$ | $\mu Wm^{-3}$ | $MJm^{-3}K^{-1}$ |
| CE | Tertiary (Cenozoic) | sand, silt and clay | 0.35 | 4.7 | 6.63 | 1.5 · | 0.7[a] | 2.95[b] |
| CR | Cretaceous | limestone with marl | 0.32 | 3.5 | 6.05 | 1.95[a] | 1.0[a] | 2.80[c] |
| J | Jurassic | claystone with silt- and sandstone | 0.2 | 4.45 | 4.05 | 2.1[a] | 1.6[a] | 3.19[c] |
| T | Triassic | silt- and sandstone | 0.5 | 8.9 | 9.85 | 2.1[a] | 1.6[a] | 2.90[c] |
| Z1 | Permian Salt | rock salt | 0.24 | 8.85 | 4.61 | 3.5[a] | 0.3[a] | 1.81[c] |
| Z2 | Permian Carbonates | gypsum and carbonate | 0.06 | 2.2 | 1.13 | 1.95[a] | 0.8[a] | 2.51[c] |
| R | Rotliegend Sediments | claystone with silt- and sandtone | 0.13 | 2.25 | 2.46 | 3[a] | 1.5[a] | 2.67[d] |
| PCV | Permo-Carboniferous Volcanics | rhyolite and andesite | 0.045 | 2.5 | 0.85 | 2.5[a] | 2.4[a] | 2.67[d] |
| PPR | Pre-Permian Rocks | strongl compacted clastics | 1.8 | 14.9 | 34.8 | 2.9[a] | 1.5[a] | 2.4[e f] |
| BG | Bohemian Granite | granite and diorite | 0.056 | 12.09 | 1.06 | 3.1[a] | 2.9[a] | 2.4[e f] |
| VUCC | Variscan Upper Crystalline Crust | granite and diorite | 1.8 | 36.7 | 34.4 | 2.8[a] | 1.3[a] | 2.5[e f] |
| UC,L | Upper Crust Laurentia | granite and diorite | 1.5 | 33.7 | 28.9 | 2.8[a] | 1.2[a] | 2.5[e f] |
| UC, A | Upper Crust Avalonia | granite and diorite | 6.6 | 34.3 | 125 | 2.9[a] | 1.3[a] | 2.5[e f] |
| UC, B | Upper Crust Baltica | granite and diorite | 13.5 | 40 | 237 | 2.75[a] | 0.9[a] | 2.5[e f] |
| LC | Lower Crust | gabbro | 8.7 | 37.2 | 165 | 2.7[a] | 0.8[a] | 2.6[e f] |
| LM | Lithospheric Mantle | peridotite | 111 | 182 | 2100 | 3.95[a] | 0.03[a] | 3.86[e g] |

[a](Scheck-Wenderoth and Maystrenko, 2013)
[b](Noack et al., 2013)
[c](Clauser, 2009)
[d](Scheck-Wenderoth et al., 2014)
[e](Maystrenko and Scheck-Wenderoth, 2013)
[f](Freymark et al., 2019)
[g](Cynn et al., 1996)

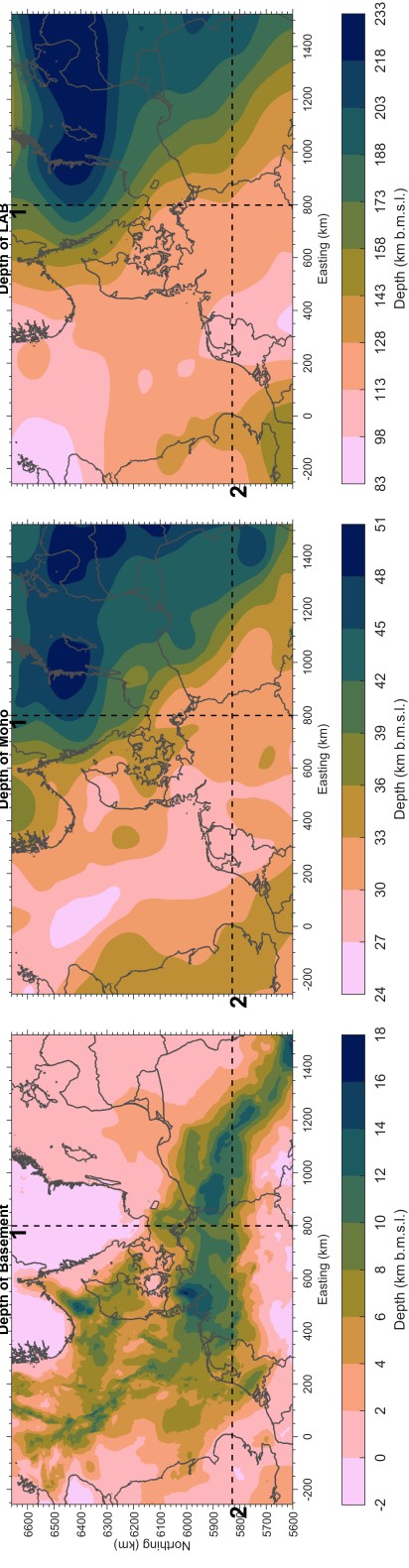

**Figure A1.** Base depth maps of the main geological discountinuity in the 3D model of the CEBS. From left to right: depth to the basement; depth to the Moho crust-mantle boundary; and depth to the Lithosphere-Asthenosphere boundary (LAB), the latter also used to imposed the bottom boundary isothermal condition. Also shown in all figures are the locations of the profiles shown in Figure A2.

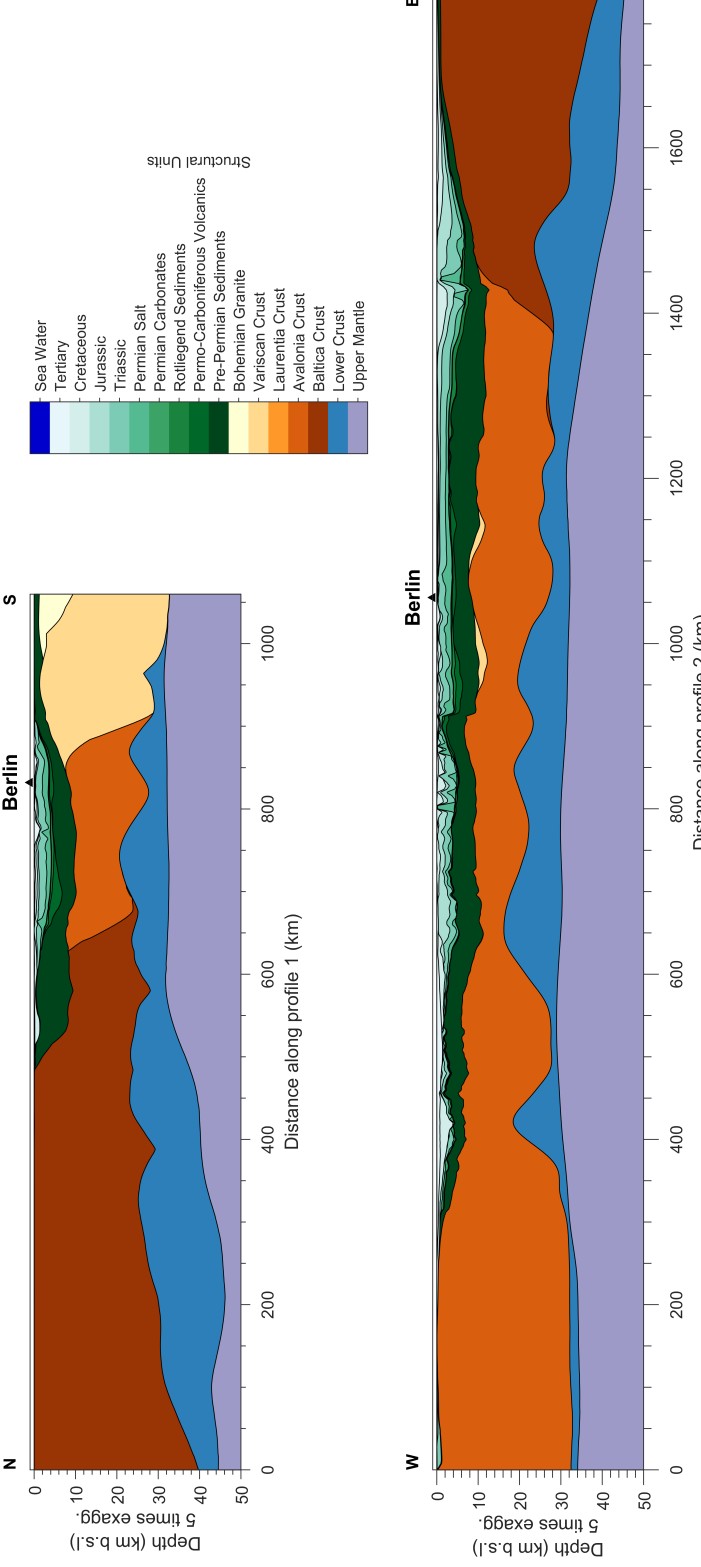

**Figure A2.** Selected geological profiles across the model illustrating the variation in space of the different structural units composing the final 3D geological model. Please, refer to Figure A1 for their locations.

*Author contributions.* All authors discussed and interpreted the presented work. DD carried out the simulations and all authors read and approved the final manuscript.

*Competing interests.* The authors declare that they have no conflict of interest.

*Acknowledgements.* The work described in this paper has received funding from the Initiative and Networking Fund of the Helmholtz Association through the project "Advanced Earth System Modelling Capacity" (ESM). The authors gratefully acknowledge the Earth System Modelling Project (ESM) for funding this work by providing computing time on the ESM partition of the supercomputer JUWELS (Jülich Supercomputing Centre, 2019) at the Jülich Supercomputing Centre (JSC) under the application 16050 entitled "Qunatitative HPC Modelling of Sedimentary Basin System." Furthermore, the authors gratefully acknowledge Dr. Uwe Mikolajewicz for providing the global ESM paleoclimate data set. Additionally, we like to thank one anonymous reviewer and Dr. Thomas Poulet for helping to improve this paper through their useful remarks and comments.

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
