# Peer review of "Effects of Transient Processes for Thermal Simulations of the Central European Basin"

_Geoscientific Model Development, 2020_

## Referee Comment (RC1) · Anonymous Referee #1 · 2 Dec 2020

I reviewed the manuscript of Degen & Cacace titled "Effects of Transient Processes for Thermal Simulations of the Central European Basin". The manuscript discusses the influence of transient processes in the subsurface temperature distribution for sedimentary basin systems. The numerical model of the Central European Basin and the sensitivity analysis of its thermal parameters are presented as a case study to evaluate their influence on the temperature field.

The work presents a new methodological approach (i.e., Reduced Basis method) to address the sensitivity analysis in thermal numerical models. This approach has been never tested in this context, and its novelty is well stated in the Introduction and method.

The results of the model and the sensitivity analysis of the thermal properties are well presented and discussed.

However, the architecture of the model should be described more in detail. The Authors reported the references of the works from which they derived the lithosphere-scale geological model of the CEBS (l. 156-157), its lateral extent, and general, brief, list of the investigated units. However, at l. 169-170, the Authors state that they are investigating the impact of the thermal properties of different chrono-stratigraphic units. At l. 193, the "Upper Crust Baltica and Avalonia" are cited and later on their thermal conductivity is reported as an "influencing thermal property". How can the Reader understand or partially figure out the extent of these units (i.e., chronostratigraphic or crustal units) and if they are spatially relevant? I suggest to: i) slightly extent the description of the geological model in the text adding some relevant information about the model architecture (for example, the thickness of the model is missing), construction, and the main units geometries, ii) add a brief lithological and spatial (i.e., maximum and minimum depths, geographical position if relevant, total volume, etc.) description of the chronostratigraphic or tectonic units in table form (this would be additionally helpful since acronyms are used to refer to these units in the figures of the results section, but these acronyms have been never explicitly stated), and iii) add a few cross sections of the geological model. I think that the impact of the obtained result would be clearer/more relevant if these data are explicitly stated in the manuscript. Especially the volume of the units could be of interest since, as the Authors are stating at l. 343, "the sensitivities of the steady-state model are mainly controlled by a combination of the volumetric contributions of the individual layers and . . .". Although the information about the model could be gathered from the literature, the work would be more complete and self-standing in my opinion.

In addition, I have a concern regarding the long-term simulation (0 ka – 255.7 Ma). This simulation is divided in 3 periods but it was not specified how the division in periods were performed. In addition, and more important, it seems that all the sedimentary

sequence was used in this model, but it is clear that, for example, the Cenozoic units were not present in the time period 75.8 Ma – 255.7 Ma since the Cenozoic started 66 Ma. Furthermore, the sedimentation of the units was different through time and, for example, the Cretaceous, Jurassic, and Triassic units were progressively not present after a certain simulation time during the time period 75.8 Ma – 255.7 Ma. I suppose that the occurrence (or not occurrence) of some units could change the final result of the sensitivity analysis considering also that the Cenozoic, Cretaceous, and Triassic units generally have the highest sensitivity indices. Was this problem considered during the simulation and eventually how was it accounted? If not, a discussion on this topic should be added.

The language is generally fluent, but some sentences are quite complex and should be simplified to achieve a better readability of the manuscript. I provided a list of these sentences and of other minor language reviews in the COMMENTS part of this letter.

Considering these points, I suggest to accept the manuscript after MINOR REVISIONS.

COMMENTS

L. 13: I am not sure if "where" is correct since the Authors are talking about what is happening in the "case nowadays". I suggest to rewrite the initial part of the sentence making it more straightforward: "This topic is especially actual since systematic efforts...".

L. 15: delete "their".

L. 19: "observations" instead of "observables"? It would fit better with the sentence at l. 20: "these datasets are spare and lacking in coverage".

L. 24-25: this sentence is not clear.

L. 25-28: I suggest to describe the factors influencing the heat distribution using a list. This would improve the readability of the sentence. In addition, I suggest to avoid

referring to "the plate" since it could be misleading. The "tectonothermal configuration of the plate" could be changed as "regional tectonothermal configuration", and the "dissipative processes within the plate" as "dissipative underground processes".

L. 31: Probably a "and" is missing. Is it "with the square root of the internal period times AND the thermal diffusivity of the plate"? As before, plate could be change with "bedrock thermal diffusivity".

L. 33-35: I suggest to split the sentence. It could improve the readability.

L. 38-40: I suggest to rephrase or split this long and complex sentence.

L. 41: "require", not "requires".

L. 53: I suggest to delete "from instance".

L. 57: If the original meaning is maintained, I suggest to change with "on the influence of the rock thermal properties". This would in better accordance with the statement at l. 61.

L. 59: as above.

L. 63-65: I suggest to rephrase this sentence. The expression "from the results of previous efforts by one of the co-authors" sounds a bit strange. I would simply put the reference as reported afterwards in the sentence. I.e: "from Degen et al. (2020)., who demonstrated . . ... ".

L. 75: check the style of references.

L. 80: "in Section 3".

L. 101: I suggest to specify the "many other methods" as done in the Introduction (l. 73). Maybe a brief description about how these other methods constructs surrogate models could be useful. This could be useful to compare with the RB method described in section 2.2.1 of the manuscript and it could highlight/strengthen the novelty of the

used approach.

L. 103: replace the comma at the end of sentence with dot.

L. 110-113: I suggest to delete this sentence since it is partially repeating previous sentences (l.91 for Sobol and Santelli methodologies; l. 85 for Wainwright et al. comparison; l. 87 for Degen et al.).

Caption 1: use the MPI-ESM abbreviation.

L. 158: A comma is missing. I.e.: "upper and lower crust, and the underlying mantle".

L. 158: I suggest to detail here the description of the model.

L. 159: I suggest to delete "to assign" or to rephrase the sentence making it more straightforward.

L. 170: this sentence would benefit from the lithological description of the units. Zechstein and Rotliegend are two chronostratigraphic units / periods that correspond to the Middle – Late Permian and Early Permian – Late Carboniferous, respectively. One could argue that they are the same units as the "Permo-Carboniferous Volcanics".

L. 172: how can we assess that the parameter correlations are negligible from the sensitivity indices? I suggest to specify it, either in brackets or in a subsequent sentence.

L: 179: I suggest to add the acronyms of the units in brackets together with their full name. In my opinion, they should be mentioned at least once when the Authors describe the result referring specifically to a figure.

L. 192-195: as above.

L. 201-206: as above.

Figure 2: what does the numbers in the X-axis mean? Just a consecutive numbering for the variable? I suggest either to remove them or to use 1 as unit for the axis. With the second option, You should get a vertical line for variable favouring the reading on

the vertical axis. If you prefer to maintain a unit different than 1, I suggest to maintain it constant among the figures (i.e., the unit on X-axis in Figure 3 is unit 2).

Caption 2: please report the reference where the Reader can find the acronyms of the different units.

Figure 3: see the comment for Figure 2.

Caption 3: see the comment for Caption 2.

L. 223: I suggest to put the accuracy value in brackets.

L. 225: I suggest to blend the short sentence together with the previous. Otherwise, replace the comma at the end of sentence with dot.

L. 230-231: these two sentences are slightly misleading. Firstly, the Authors say that "the results are the same for all accuracies tested", but then they state that there are differences among the different accuracies for parameters with low sensitivity. It is clear that the impact of the accuracies is minor since the sensitivity of parameters is low, but the first statement goes in the opposite direction. I suggest to rephrase the sentence at l. 230 describing more in detail the results shown in the figure. A better description of the results will avoid any misinterpretation.

Figure 4: see the comment for Figure 2.

Caption 4: see the comment for Caption 2.

L. 238-239: I suggest to split or rephrase this sentence to increase its readability. The construction of the sentence is quite complex.

L. 240-242: as above.

Figure 5: see the comment for Figure 2.

Caption 5: see the comment for Caption 2.

L. 253-259: as comment for L. 179.

[Figure]

Figure 7: the title on the Y-axis is missing.

Caption 7: see the comment for Caption 2.

L. 280: I suggest to use "a time-variable scaling factor". The term "increasing" could be misleading since it is stated that the uncertainties in the temperatures should decrease with time.

L. 293: as comment for L. 179.

L. 296: what does the "the errors in their sensitivities" mean? How can the Reader assess this error? Is it the accuracy discussed in section 3.2 or another parameter? I suggest to specify it and eventually restate the accuracy value of the model.

Caption 11: see the comment for Caption 2.

L. 303-305: why did the Author choose to perform the sensitivity analysis in 4 different periods? Does the segmentation in periods have a geological meaning? These aspects should be specified. In addition, I suggest to start the bullet list from the period 0 – 22.8 ka.

L. 307-308: as comment for L. 179.

L. 321-324: as comment for L. 179.

L. 348: I suggest to put a set of representative values or a reference for validating the sentence "This is caused by the higher radiogenic heat production of the latter rocks". The same suggestion can be referred to the thermal conductivity of Zechstein (l. 351) and of the lithospheric mantle (l. 352).

L. 353: please specify the percentage of the lithospheric mantle volume with respect of the total volume of the model.

L. 385: check the "to imposed" and eventually rephrase the sentence.

L. 393: "make improve" is not correct in my opinion, I would just keep "improve".

L. 430: "an additional"
* * *

---

## Referee Comment (RC2) · Thomas Poulet (Referee) · 9 Dec 2020

This paper investigates the effects of paleoclimate variation on conductive heat transfer in sedimentary basins. The authors demonstrate an impact that is too often neglected and quantify robustly the effects of surface temperature evolution on the subsurface at various depths, depending on the timescales considered, through a global sensitivity analysis (GSA). This work is very relevant for the geoscientific modelling community and the workflow presented shows clearly the importance of GSA - possible thanks to efficient surrogate models – to not only identify the parameters of importance but also their correlations. I found the study particularly well adapted for the journal as it

presents a novel modelling workflow to alleviate many of the issues arising from local sensitivity analyses and manual calibrations. Overall, the manuscript is well structured and the message really clear. In my opinion, the impact of this works certainly warrants its publication in GMD, even though I recommend quite a few minor revisions to clarify some points detailed below and improve the manuscript.

General comments:

\* All figures showing indices (e.g fig.2,3,4,5,7,11,12. . .) would make more sense plotted as (2 colour pairs of) histograms since the x-axis is not continuous but represents the discrete parameters. The legends of those figures should also point the reader to a table (or the new figure) describing the acronyms of the geological units.

\* Across the whole manuscript, the main text should be more self-contained in the sense that description and results of the figures should appear explicitly in the text as well. This is currently done for figs.1,2,7,11 but not the others. The text only mentions that "the results are shown in fig." 3 (L.187), 4(L.200), 10 (L.283) and makes implicit references to figs.5,6,8,9,13,14,15. I don't think fig.12 is even referenced in the text.

Specific comments:

\* A figure is missing to display the geological model, so that the reader can understand better the (implicit) links between the geological units' names (i.e ages) and depth, which is an important aspect of the results. I understand and agree with the approach of not focusing too much on the geologic model (which is quite irrelevant for this study) but a minimum must be mentioned including the number of units, which could maybe be listed in some sort of order of depth (of the centre of mass?) as this is the most relevant aspect.

\* Please check equations 1 and 2 which seem to have a few problems and confirm these typos don't affect any of the results

|— Eq. 1, elements to check/correct:

[Figure]

|—— Minus sign for the diffusion term

|—— l_refˆ2 in the source term

|— Eq. 2:

|—— Minus sign for the diffusion term

|—— l_refˆ2 in t_ref

|—— S_ref instead of S_(s,ref) in the second term (heat production)

|—— S_(s,ref) seems to include C_p and is therefore not necessarily the standard definition of "specific radiogenic heat production", so please specify your definition (and adapt the name if needed).

|—— missing "\partial" in front of time

* The second part of the introduction (L.48-80) needs some touches to improve the reading flow. The mention of all components of the paper could benefit from adverbs and some reordering to emphasise the logic in which the info is introduced: first "why" some work needs to be done, then "what" are the goals of the paper, and finally "how" you're going about it. For all points, the justification of the work should indeed appear before the mention of the elements themselves, instead of afterwards (which weakens the points by following more of a "report" format, e.g. L.57-59 before L.55; L.61-64 before L.60; L.71-73 before L.70). More explicit logical links will help transform the current impression of a listing of elements ("the main goal of this study" L.48, "we will describe and discuss" L.50, "we will demonstrate" L.52, "the aim of the study" L.54, "in this paper we present" L.60, "our case study is" L.74), which currently leaves the task of connecting them to the reader.

* L.58 mentions that paleoclimate effects on subsurface heat have only been looked at in 1D and provides a good but old reference from 1984. The impact of paleoclimate on deep heat flux is indeed often underestimated, but more recent work should be

mentioned as well (and again as a justification for the work beforehand). See for instance (Dentzer et al, 2016, http://doi.org/10.1016/j.geothermics.2016.01.006) and all references therein.

* Sec. 2.1, a short sentence would be welcome to actually explain/summarise what the Sobol sensitivity analysis and Saltelli sampling routines are.

* Sec. 2.3, briefly mention what kind of constraints are used to calibrate that model, which initial conditions are used.

* Sec. 2.3, which absolute time period are the relative time steps 0-26ka supposed to represent? (Reader is only learning l.248 that the number 26k results from the reconstructed paleotemperatures available. This info should appear with the first mention of "26k")

* L.172, the contributions being "negligible" imply some relative thresholds that are not specified. An extra sentence would be nice to comment on absolute and relative thresholds the authors used for all indices in this study. For instance, are you choosing three(x2) parameters L.190 because you consider 0.1 to be a good threshold?

* L.174, the number is not 5 but actually 5x2=10 since later examples show that you're not picking geological units but parameters which are not necessarily in the same units. Please check the whole text for consistency (e.g. "three" L.190,...)

* L.205, this information should appear at the beginning of the section (as mentioned above, justifications should appear before descriptions): from what I understood, you want to end up with a manageable number (arbitrarily 8) of most sensitive parameters overall but it would be too expensive to run a GSA with all parameters at once, so you break down the problem to first identify the most sensitive in the sediments, then in other areas, and then pick those parameters for further study.

* L.212 "higher accuracy than typical temperature measurements" -> what does that mean exactly? You mentioned that you're not solving for temperature but heat, so how

does 5e-4 accuracy on a model translate into temperature measurement precision? (Similarly L.222, why 4e-3 rather than 5e-3 to relax by one order of magnitude?).

* Sec. 3.3.2, I only understood what was being simulated in this section after analysing fig.9 (mentioned L.273). Indeed, Fig.7 (mentioned L.251) shows an average initial temperature around -5C and a final temperature around 8C, which I could not instantly reconcile with applying a Dirichlet boundary condition of 1.6C mentioned L.246.

* Sec. 3.3.2, I also don't get the point of fitting the average temperature with a 4th order polynomial. Why not use directly the average temperature itself discretised at your transient time step? Why do you need a smoother version? As for the smoothing, it's impossible to judge a fitting quality without any mention the metric used to assess the impact. I agree (L.277) that the 5th order polynomial doesn't significantly improve the fit compared to the 4th order visually, yet the fit remains rather poor in my eyes (rough estimate of max(deltaT)∼3C) and the selection of the best fitting function is a moot point without specifying both the metric to assess the fit and the cost of using a higher order polynomial/smoother fit.

* L.280, what is this "scaling factor" and what is it applying to? (I can only start guessing after seeing fig.10, describe it explicitly in the text.)

* L.285, the mention of "glaciation times" comes out of the blue and should be introduced.

* Sec 3.3.3 looks a bit odd at first sight as it seems to draw an opposite conclusion to the paper itself, with the transient boundary conditions adding no value over constant ones (the main properties showing "no significant changes" L.292 and the others being "insignificant" L.293). Please manage the delivery of this message.

* L.416-419, I don't quite get the need to deduce the obvious, that heat moves upwards in this setting, nor the less obvious conclusion of why sediments at the uppermost part have therefore a more prominent influence.

* L.443-446, this information would be better suited in section 3 to justify the approach when presenting it.

* Sec. 5, the conclusion needs some polishing. It looks a bit too much like a series of collated dot points with a succession of short sentences (e.g. last paragraph). The paragraph breakdown is awkward with two of them containing a single sentence (L.457, 464). Emphasise more the causal relationships by introducing some segways or logical links, and please amend the abrupt finish to leave the readers on a more impactful last sentence.

Technical corrections

* L.56 & L.58 "influence on [the calibration of] thermal properties"

* Sentence L.70-74 could be easier to read if reordered –> "In this study, we make use of the RB method . . . since it allows" (l.71) "the retrieval of the entire state variable (i.e. temperature)" (l.73), "in contrast to other statistical methods. . ." (l.71-73)

* L.96 add missing words: [It is] "worth mentioning. . ."

* The last two sentences of sec. 2.1 (L.108-111) should come L.93, after the mention of Sobol GSA (L.89-93) but before the description of the cost function (L.94-108)

* (For Eq.(1) and (2), you might want to add a note to point out that the Laplace operator applies with respect to the normalised space. I can see why you wrote those equations this way, to avoid defining all dimensionless variables and parameters, but since you're only using symbols carrying physical dimensions the Laplace operator is slightly misleading, strictly speaking. This pedantic comment is optional.)

* Legend of fig.1, not clear (at this point) if the times (0, 13, 26ka) refer to absolute dates (in which case they would be better displayed in inverse order) or are in chronological from an unspecified reference for 0ka.

* L.150,151 What are T31 and GR30? Specify a bit the nature of those models and/or

add references.

* L.157, mentioning the depth of the LAB would be informative.

* Remove comma L.158 "throughout, the entire paper"

* L.174, why picking 5 parameters (rather than 4 or 6. . .)?

* Remove comma L.208 "the investigation[s] carried out so far, have enabled.."

* Legend of fig.2: Missing mention of the horizontal black line (separation of radiogenic heat and thermal conductivity parameters?) and two boxes (first five) in the figure legend.

* L.224 "However, with a significantly lower computational cost" (sentence segment, no verb. . .)

* L.224-227: a bit confusing, please rephrase with something along the lines of "Despite potentially introducing additional error sources with a relaxed tolerance, this accuracy drop can actually be considered insignificant. Indeed, sensitivity analyses are based on. . .. Since all simulations are . . . see Fig.5."

* L.230: is "however" the correct logical link?

* Sentence L.239-241 As an introduction sentence to the section, keep it at present tense, not past/conditional tense ("having been able". . . "could")

* Fig.6 branch2 should mention "with paleoclimate"

* L.251 "Fig. 7 compares the sensitivities of the thermal properties for the steady-state and transient system [with the selected initial and boundary conditions]". One would indeed expect the results of fig.7 to vary with different initial conditions and/or transient boundary conditions.

* L.267 "ar[i]sing"

* L.275, Eq(3), add something like $T\_top\ (t)=$âŃŕ to make it a proper equation

* L.305 "a second discussion point [is]"

* Fig.12, the collage is appropriate, but all fonts need to be slightly increased accordingly

* L.319-320: "very similar" and "showing some major changes" are puzzling/contradictory in the same sentence

* L.343 not sure I fully understand the wording "combination of the volumetric contributions of the individual layers and their thermal properties". Thermal conductivity and radiogenic heat production are both volumetric properties. What does "and their" imply rather than " ' " ("individual layers' thermal properties")?

* L.355 "as apparent by the [insignificant] difference between..."

* L. 358: not sure why you put "base" between inverted commas(?)

* L.398, the wording "similar but not identical" doesn't do justice to the importance of this difference.

* L.402 "Fig. 6 branch 1 [b]"

* L.429 "a[n] additional"

* L.440 "Only the consideration of these variations could enable us.."

* L.448 "w[h]ere"

* L.450 "fourth" <- cardinal number needed (not ordinal)

* L.455: might want to rephrase "since the temperature diffuses over time towards the bottom of the model" as the heat moves upwards and the cold top Dirichlet boundary condition leads to a perceived propagation of a cold front downwards...

* L.474, reformulate sentence "Using the finite element method the here presented analyses computationally prohibitive, only the utilization of a surrogate model allows the execution of these analyses."

---

## Author Comment (AC1) · 6 Jan 2021

First of all, we would like to thank RC1 for all points raised which helped us to improve not only the readability of the manuscript but its scientific merit. In the attachment to this post, you will find a first file (RC1_revision.pdf) where we detail our point by point revision, each RC1's comments is followed by our answers (highlighted in a different colour). We also attached three figures (two about the details of the geological model and one of a table summarizing the properties plus geological information) that we added as Appendix material to the revised version of our manuscript as requested both by RC1 and RC2. Please also refer to the revised manuscript in this regard.

[Figure]

Please also note the supplement to this comment:
https://gmd.copernicus.org/preprints/gmd-2020-204/gmd-2020-204-AC1-
supplement.pdf

––––––––––––––––––––––––––––––––

[Figure]

**Fig. 1.** base maps of the main geological units in the 3D CEBS geological model

[Figure]

**Fig. 2.** selected geoloigcal profiles across the 3D CEBS model

**Table A1.** Geometrical and rock phsical properties of the different units integrated in the 3D structural model. Symbols listed: h (av)=average unit thickness, h (max)=maximum unit thickness, $\lambda$=thermal conductivity, $H$=heat production rate, $\rho$=density, and $c_p$=specific heat capacity. The unit volume has been computed based on the average thickness of each unit.

| Acronym | Layer | main lithology | h (av) | h (max) | volume | $\lambda$ | $H$ | $\rho \times c_p$ |
|---|---|---|---|---|---|---|---|---|
| - | - | - | $km$ | $km$ | $10^5 km^3$ | $Wm^{-1}K^{-1}$ | $\mu Wm^{-3}$ | $MJm^{-3}K^{-1}$ |
| CE | Tertiary (Cenozoic) | sand, silt and clay | 0.35 | 4.7 | 6.63 | 1.5[a]. | 0.7[?] | 2.95[b] |
| CR | Cretaceous | limestone with marl | 0.32 | 3.5 | 6.05 | 1.95[?] | 1.0[?] | 2.80[?] |
| J | Jurassic | claystone with silt- and sandstone | 0.2 | 4.45 | 4.05 | 2.1[?] | 1.6[?] | 3.19[?] |
| T | Triassic | silt- and sandstone | 0.5 | 8.9 | 9.85 | 2.1[?] | 1.6[?] | 2.90[?] |
| Z1 | Permian Salt | rock salt | 0.24 | 8.85 | 4.61 | 3.5[?] | 0.3[?] | 1.81[?] |
| Z2 | Permian Carbonates | gypsum and carbonate | 0.06 | 2.2 | 1.13 | 1.95[?] | 0.8[?] | 2.51[c] |
| R | Rotliegend Sediments | claystone with silt- and sandtone | 0.13 | 2.25 | 2.46 | 3[?] | 1.5[?] | 2.67[d] |
| PCV | Permo-Carboniferous Volcanics | rhyolite and andesite | 0.045 | 2.5 | 0.85 | 2.5[?] | 2.4[?] | 2.67[?] |
| PPR | Pre-Permian Rocks | strongl compacted clastics | 1.8 | 14.9 | 34.8 | 2.9[?] | 1.5[?] | 2.4[e f] |
| BG | Bohemian Granite | granite and diorite | 0.056 | 12.09 | 1.06 | 3.1[?] | 2.9[?] | 2.4[? ?] |
| VUCC | Variscan Upper Crystalline Crust | granite and diorite | 1.8 | 36.7 | 34.4 | 2.8[?] | 1.3[?] | 2.5[? ?] |
| UC,L | Upper Crust Laurentia | granite and diorite | 1.5 | 33.7 | 28.9 | 2.8[?] | 1.2[?] | 2.5[? ?] |
| UC, A | Upper Crust Avalonia | granite and diorite | 6.6 | 34.3 | 125 | 2.9[?] | 1.3[?] | 2.5[? ?] |
| UC, B | Upper Crust Baltica | granite and diorite | 13.5 | 40 | 237 | 2.75[?] | 0.9[?] | 2.5[? ?] |
| LC | Lower Crust | gabbro | 8.7 | 37.2 | 165 | 2.7[?] | 0.8[?] | 2.6[? ?] |
| LM | Lithospheric Mantle | peridotite | 111 | 182 | 2100 | 3.95[?] | 0.03[?] | 3.86[? ? g] |

[a](?)
[b](?)
[c](?)
[d](?)
[e](?)
[f](?)
[g](?)

**Fig. 3.** Table geological and properties

**Supplement:**

Point by point answers to the comments from reviewer#1 to the manuscript entitled "Effects of Transient Processes for Thermal Simulations of the Central European Basin" by Degen and Cacace.

Anonymous Referee #1

I reviewed the manuscript of Degen & Cacace titled "Effects of Transient Processes for Thermal Simulations of the Central European Basin". The manuscript discusses the influence of transient processes in the subsurface temperature distribution for sedimentary basin systems. The numerical model of the Central European Basin and the sensitivity analysis of its thermal parameters are presented as a case study to evaluate their influence on the temperature field.

The work presents a new methodological approach (i.e., Reduced Basis method) to address the sensitivity analysis in thermal numerical models. This approach has been never tested in this context, and its novelty is well stated in the Introduction and method. The results of the model and the sensitivity analysis of the thermal properties are well presented and discussed. However, the architecture of the model should be described more in detail. The Authors reported the references of the works from which they derived the lithosphere-scale geological model of the CEBS (l. 156-157), its lateral extent, and general, brief, list of the investigated units. However, at l. 169-170, the Authors state that they are investigating the impact of the thermal properties of different chrono-stratigraphic units. At l. 193, the "Upper Crust Baltica and Avalonia" are cited and later on their thermal conductivity is reported as an "influencing thermal property". How can the Reader understand or partially figure out the extent of these units (i.e., chronostratigraphic or crustal units) and if they are spatially relevant? I suggest to: i) slightly extent the description of the geological model in the text adding some relevant information about the model architecture (for example, the thickness of the model is missing), construction, and the main units geometries, ii) add a brief lithological and spatial (i.e., maximum and minimum depths, geographical position if relevant, total volume, etc.) description of the chronostratigraphic or tectonic units in table form (this would be additionally helpful since acronyms are used to refer to these units in the figures of the results section, but these acronyms have been never explicitly stated), and iii) add a few cross sections of the geological model. I think that the impact of the obtained

result would be clearer/more relevant if these data are explicitly stated in the manuscript. Especially the volume of the units could be of interest since, as the Authors are stating at l. 343, "the sensitivities of the steady-state model are mainly controlled by a combination of the volumetric contributions of the individual layers and … ". Although the information about the model could be gathered from the literature, the work would be more complete and self-standing in my opinion.

In our original submission we did not provide details of the 3D structural model used as input for the sensitivity analysis. This was done to keep the paper light and easy to read/follow and not to "distract" the reader from the main topic of the study, that is, the description of a novel methodology to address global sensitivity analysis for thermal model in sedimentary basins. This is also manifested in our decision to target GMD as a journal where to publish our results, given the novel theoretical and computational aspects that make up the bulk of the study. In this regard, we consider the application to the CEBS as a "proof of concept" rather than the core of the study as also detailed in our final discussion to the manuscript.

This said, given also a similar comment from reviewer#2, we have integrated all requested information on the 3D structural model by expanding its description in the main text as well as in the form of two additional figures and a descriptive table in the revised version of the paper.

In addition, I have a concern regarding the long-term simulation (0 ka – 255.7 Ma). This simulation is divided in 3 periods but it was not specified how the division in periods were performed. In addition, and more important, it seems that all the sedimentary sequence was used in this model, but it is clear that, for example, the Cenozoic units were not present in the time period 75.8 Ma – 255.7 Ma since the Cenozoic started 66 Ma. Furthermore, the sedimentation of the units was different through time and, for example, the Cretaceous, Jurassic, and Triassic units were progressively not present after a certain simulation time during the time period 75.8 Ma – 255.7 Ma. I suppose that the occurrence (or not occurrence) of some units could change the final result of the sensitivity analysis considering also that the Cenozoic, Cretaceous, and Triassic units generally have the highest sensitivity

indices. Was this problem considered during the simulation and eventually how was it accounted? If not, a discussion on this topic should be added.

We have opted for subdividing the total simulation time into three main periods because we wanted to being able to isolate short-term, mid-term and long-term effects on the model results. This said, we added some extensive explanatory text to the original manuscript (l. 375-377).

Regarding the geological model, the reviewer is right in that we have made use of a single model throughout the whole paper. The reason behind our choice stems from our aim to provide and test our approach to isolate and quantify the non-linear and therefore superposed effects induced by considering a change in the system dynamics (from a steady- to a transient state), an aspect that still provides difficult in the context of basin-wide thermal modelling and that as such has been never quantified so far. To allow variations in the input geological model over time (and questions would then arise on how to best represent those variations if only based on the present-day information available) would have increased the parameter space in a rather unpredictable manner, thereby resulting in an over-parameterised problem. All these aspects would have hindered a proper assessment of the model outcomes and of the consequences of the physical processes that we targeted in the study. In order to provide some clarification on this aspect, we have added some explanatory text to the original manuscript following the same reasoning as outlined above (l. 533-542).

The language is generally fluent, but some sentences are quite complex and should be simplified to achieve a better readability of the manuscript. I provided a list of these sentences and of other minor language reviews in the COMMENTS part of this letter.

We have acknowledged all comments from reviewer#1 and modified the original text accordingly.

Considering these points, I suggest to accept the manuscript after MINOR REVISIONS.

COMMENTS

L. 13: I am not sure if "where" is correct since the Authors are talking about what is happening in the "case nowadays". I suggest to rewrite the initial part of the sentence making it more straightforward: "This topic is especially actual since systematic efforts … ".

The sentence was rephrased.

L. 15: delete "their".

Has been deleted.

L. 19: "observations" instead of "observables"? It would fit better with the sentence at l. 20: "these datasets are spare and lacking in coverage".

Has been changed.

L. 24-25: this sentence is not clear.

We have rephrased the sentence to improve its readability.

L. 25-28: I suggest to describe the factors influencing the heat distribution using a list. This would improve the readability of the sentence. In addition, I suggest to avoid referring to "the plate" since it could be misleading. The "tectonothermal configuration of the plate" could be changed as "regional tectonothermal configuration", and the "dissipative processes within the plate" as "dissipative underground processes".

All points have been addressed in the revised version of the manuscript.

L. 31: Probably a "and" is missing. Is it "with the square root of the internal period times AND the thermal diffusivity of the plate"? As before, plate could be change with "bedrock thermal diffusivity".

The sentence was correct. There, is no missing "and".

L. 33-35: I suggest to split the sentence. It could improve the readability.

The sentence was split.

L. 38-40: I suggest to rephrase or split this long and complex sentence.

The sentence was split.

L. 41: "require", not "requires".

This has been changed accordingly.

L. 53: I suggest to delete "from instance".

Done.

L. 57: If the original meaning is maintained, I suggest to change with "on the influence

of the rock thermal properties". This would in better accordance with the statement at l. 61.

The original meaning would change with the reformulation. For this reason we kept the original formulation.

L. 59: as above.

As above.

L. 63-65: I suggest to rephrase this sentence. The expression "from the results of previous efforts by one of the co-authors" sounds a bit strange. I would simply put the reference as reported afterwards in the sentence. I.e: "from Degen et al. (2020)., who demonstrated ... ".

The sentence has been rephrased.

L. 75: check the style of references.

The style has been corrected.

L. 80: "in Section 3".

Has been corrected.

L. 101: I suggest to specify the "many other methods" as done in the Introduction (l. 73). Maybe a brief description about how these other methods constructs surrogate models could be useful. This could be useful to compare with the RB method described in section 2.2.1 of the manuscript and it could highlight/strengthen the novelty of the used approach.

We have followed the reviewer's suggestion and provide a better description of the referred methods. In doing so, we did not further enter detailed descriptions of other surrogate methods since they do not fit with the main topic addressed in this study given their focus on the observation space (as we stated in the paper). Indeed, in our study, we focus on the entire temperature distribution thereby making other kinds of surrogate model unsuitable.

L. 103: replace the comma at the end of sentence with dot.

Has been addressed.

L. 110-113: I suggest to delete this sentence since it is partially repeating previous sentences (l.91 for Sobol and Santelli methodologies; l. 85 for Wainwright et al. comparison;

l. 87 for Degen et al.).

The sentence was been moved to another location as suggested by reviewer#2.

Caption 1: use the MPI-ESM abbreviation.

Has been changed

L. 158: A comma is missing. I.e.: "upper and lower crust, and the underlying mantle".

Has been changed

L. 158: I suggest to detail here the description of the model.

Some informative text has been added, as well as additional materials (in the form of a descriptive table plus base maps of major horizons in the Appendix to the revised text).

L. 159: I suggest to delete "to assign" or to rephrase the sentence making it more straightforward.

Has been deleted.

L. 170: this sentence would benefit from the lithological description of the units. Zechstein and Rotliegend are two chronostratigraphic units / periods that correspond to the Middle – Late Permian and Early Permian – Late Carboniferous, respectively. One could argue that they are the same units as the "Permo-Carboniferous Volcanics".

We have added a first order lithological description of the different units (consider the main lithology as being representative of the bulk volume of the respective layer) in the Table that can be found in the Appendix to the revised manuscript.

L. 172: how can we assess that the parameter correlations are negligible from the sensitivity indices? I suggest to specify it, either in brackets or in a subsequent sentence.

Parameter correlations can be asset through the difference between first- and total-order indices. An explanation has been added to the revised paper.

L: 179: I suggest to add the acronyms of the units in brackets together with their full name. In my opinion, they should be mentioned at least once when the Authors describe the result referring specifically to a figure.

The acronyms have been added to within a table in the Appendix to the revised manuscript.

L. 192-195: as above.

As above

L. 201-206: as above.

As above

Figure 2: what does the numbers in the X-axis mean? Just a consecutive numbering for the variable? I suggest either to remove them or to use 1 as unit for the axis. With the second option, You should get a vertical line for variable favouring the reading on the vertical axis. If you prefer to maintain a unit different than 1, I suggest to maintain it constant among the figures (i.e., the unit on X-axis in Figure 3 is unit 2).

The plot has been modified to better illustrate the meaning of the x-axis as a discontinuous unit.

Caption 2: please report the reference where the Reader can find the acronyms of the different units.

A reference to the table containing the acronyms has been added.

Figure 3: see the comment for Figure 2.

The plot has been modified to better illustrate the meaning of the x-axis as a discontinuous unit.

Caption 3: see the comment for Caption 2.

A reference to the table containing the acronyms has been added.

L. 223: I suggest to put the accuracy value in brackets.

The accuracy has been put in brackets.

L. 225: I suggest to blend the short sentence together with the previous. Otherwise, replace the comma at the end of sentence with dot.

The sentence has been modified.

L. 230-231: these two sentences are slightly misleading. Firstly, the Authors say that "the results are the same for all accuracies tested", but then they state that there are differences among the different accuracies for parameters with low sensitivity. It is clear that the impact of the accuracies is minor since the sensitivity of parameters is low, but the first statement goes in the opposite direction. I suggest to rephrase the sentence at l. 230 describing more in detail the results shown in the figure. A better description of the results will avoid any misinterpretation.

The sentences have been reformulated to improve their readability.

Figure 4: see the comment for Figure 2.

The plot has been modified to better illustrate the meaning of the x-axis as a discontinuous unit.

Caption 4: see the comment for Caption 2.

A reference to the table containing the acronyms has been added.

L. 238-239: I suggest to split or rephrase this sentence to increase its readability. The construction of the sentence is quite complex.

The sentence has been modified.

L. 240-242: as above.

The sentence has been modified.

Figure 5: see the comment for Figure 2.

The plot has been modified to better illustrate the meaning of the x-axis as a discontinuous unit.

Caption 5: see the comment for Caption 2.

A reference to the table containing the acronyms has been added.

L. 253-259: as comment for L. 179.

The acronyms have been added to Table A1.

Figure 7: the title on the Y-axis is missing.

The title has been added.

Caption 7: see the comment for Caption 2.

The acronyms have been added to Table A1.

L. 280: I suggest to use "a time-variable scaling factor". The term "increasing" could be misleading since it is stated that the uncertainties in the temperatures should decrease with time.

The formulation has been adopted.

L. 293: as comment for L. 179.

The acronyms have been added to Table A1.

L. 296: what does the "the errors in their sensitivities" mean? How can the Reader

assess this error? Is it the accuracy discussed in section 3.2 or another parameter? I suggest

to specify it and eventually restate the accuracy value of the model.

An explanation has been added to the paper.

Caption 11: see the comment for Caption 2.

A reference to the table containing the acronyms has been added.

L. 303-305: why did the Author choose to perform the sensitivity analysis in 4 different periods? Does the segmentation in periods have a geological meaning? These aspects should be specified. In addition, I suggest to start the bullet list from the period 0 – 22.8 ka.

A bullet list has been inserted and we also provide an explanation of the subdivision made (see also our comments to the reviewer's general remarks).

L. 307-308: as comment for L. 179.

The acronyms have been added to the table in the Appendix of the revised manuscript.

L. 321-324: as comment for L. 179.

The acronyms have been added to the table in the Appendix of the revised manuscript.

L. 348: I suggest to put a set of representative values or a reference for validating the sentence "This is caused by the higher radiogenic heat production of the latter rocks". The same suggestion can be referred to the thermal conductivity of Zechstein (l. 351) and of the lithospheric mantle (l. 352).

Crustal rocks have a granitoid prevailing lithological composition, thus their higher than sedimentary rocks heat production rates. Zechstein is a salt rock, while the mantle is considered as peridotite enriched, therefore their higher thermal conductivity.

L. 353: please specify the percentage of the lithospheric mantle volume with respect of the total volume of the model.

Information about the volume of the respective volume for each specific unit has been added in a table, which we have added to the revised version of the manuscript. By considering the values provided in the table, we derived that the mantle makes up around 76% of the total volume. We have added this information in the text.

L. 385: check the "to imposed" and eventually rephrase the sentence.

Has been corrected.

L. 393: "make improve" is not correct in my opinion, I would just keep "improve".

Has been corrected.

L. 430: "an additional"

Has been corrected.

---

## Author Comment (AC2) · 6 Jan 2021

We would like to thank RC2 (Dr. Thomas Poulet) for his comments which improved the scientific merit of our study. In the attachment to this post, you will find a file (RC2_revision.pdf) where we detail our point by point revision, each RC2's comments is followed by our answers (highlighted in a different colour). As also for our post to RC1's comments, we also attached additional informative materials as figures. There are three figures in total (two about the details of the geological model and one of a table summarizing the properties plus geological information) that we added as Appendix material to the revised version of our manuscript as requested both by RC1

and RC2. Please also refer to the revised manuscript in this regard.

Please also note the supplement to this comment:
https://gmd.copernicus.org/preprints/gmd-2020-204/gmd-2020-204-AC2-
supplement.pdf

———————————————————

[Figure]

**Fig. 1.** base maps of the main geological units in the 3D CEBS geological model

[Figure]

**Fig. 2.** selected geological profiles across the 3D CEBS model

Table A1. Geometrical and rock phsical properties of the different units integrated in the 3D structural model. Symbols listed: h (av)=average unit thickness, h (max)=maximum unit thickness, $\lambda$=thermal conductivity, $H$=heat production rate, $\rho$=density, and $c_p$=specific heat capacity. The unit volume has been computed based on the average thickness of each unit.

| Acronym | Layer | main lithology | h (av) | h (max) | volume | $\lambda$ | $H$ | $\rho \times c_p$ |
|---|---|---|---|---|---|---|---|---|
| - | - | - | $km$ | $km$ | $10^5 km^3$ | $Wm^{-1}K^{-1}$ | $\mu Wm^{-3}$ | $MJm^{-3}K^{-1}$ |
| CE | Tertiary (Cenozoic) | sand, silt and clay | 0.35 | 4.7 | 6.63 | $1.5^a.$ | $0.7^{??}$ | $2.95^b$ |
| CR | Cretaceous | limestone with marl | 0.32 | 3.5 | 6.05 | $1.95^{??}$ | $1.0^{??}$ | $2.80^{??}$ |
| J | Jurassic | claystone with silt- and sandstone | 0.2 | 4.45 | 4.05 | $2.1^{??}$ | $1.6^{??}$ | $3.19^{??}$ |
| T | Triassic | silt- and sandstone | 0.5 | 8.9 | 9.85 | $2.1^{??}$ | $1.6^{??}$ | $2.90^{??}$ |
| Z1 | Permian Salt | rock salt | 0.24 | 8.85 | 4.61 | $3.5^{??}$ | $0.3^{??}$ | $1.81^{??}$ |
| Z2 | Permian Carbonates | gypsum and carbonate | 0.06 | 2.2 | 1.13 | $1.95^{??}$ | $0.8^{??}$ | $2.51^c$ |
| R | Rotliegend Sediments | claystone with silt- and sandtone | 0.13 | 2.25 | 2.46 | $3^{??}$ | $1.5^{??}$ | $2.67^d$ |
| PCV | Permo-Carboniferous Volcanics | rhyolite and andesite | 0.045 | 2.5 | 0.85 | $2.5^{??}$ | $2.4^{??}$ | $2.67^{??}$ |
| PPR | Pre-Permian Rocks | strongl compacted clastics | 1.8 | 14.9 | 34.8 | $2.9^{??}$ | $1.5^{??}$ | $2.4^{e\ f}$ |
| BG | Bohemian Granite | granite and diorite | 0.056 | 12.09 | 1.06 | $3.1^{??}$ | $2.9^{??}$ | $2.4^{??\ ??}$ |
| VUCC | Variscan Upper Crystalline Crust | granite and diorite | 1.8 | 36.7 | 34.4 | $2.8^{??}$ | $1.3^{??}$ | $2.5^{??\ ??}$ |
| UC,L | Upper Crust Laurentia | granite and diorite | 1.5 | 33.7 | 28.9 | $2.8^{??}$ | $1.2^{??}$ | $2.5^{??\ ??}$ |
| UC, A | Upper Crust Avalonia | granite and diorite | 6.6 | 34.3 | 125 | $2.9^{??}$ | $1.3^{??}$ | $2.5^{??\ ??}$ |
| UC, B | Upper Crust Baltica | granite and diorite | 13.5 | 40 | 237 | $2.75^{??}$ | $0.9^{??}$ | $2.5^{??\ ??}$ |
| LC | Lower Crust | gabbro | 8.7 | 37.2 | 165 | $2.7^{??}$ | $0.8^{??}$ | $2.6^{??\ ??}$ |
| LM | Lithospheric Mantle | peridotite | 111 | 182 | 2100 | $3.95^{??}$ | $0.03^{??}$ | $3.86^{??\ g}$ |

[a](?)
[b](?)
[c](?)
[d](?)
[e](?)
[f](?)
[g](?)

**Fig. 3.** Table geological and properties

**Supplement:**

Point by point answers to the comments from reviewer#2 to the manuscript entitled "Effects of Transient Processes for Thermal Simulations of the Central European Basin" by Degen and Cacace.

Thomas Poulet (Referee)

thomas.poulet@csiro.au

This paper investigates the effects of paleoclimate variation on conductive heat transfer in sedimentary basins. The authors demonstrate an impact that is too often neglected and quantify robustly the effects of surface temperature evolution on the subsurface at various depths, depending on the timescales considered, through a global sensitivity analysis (GSA). This work is very relevant for the geoscientific modelling community and the workflow presented shows clearly the importance of GSA - possible thanks to efficient surrogate models – to not only identify the parameters of importance but also their correlations. I found the study particularly well adapted for the journal as it presents a novel modelling workflow to alleviate many of the issues arising from local sensitivity analyses and manual calibrations. Overall, the manuscript is well structured and the message really clear. In my opinion, the impact of this works certainly warrants its publication in GMD, even though I recommend quite a few minor revisions to clarify some points detailed below and improve the manuscript.

General comments:

* All figures showing indices (e.g fig.2,3,4,5,7,11,12: : :) would make more sense plotted as (2 colour pairs of) histograms since the x-axis is not continuous but represents the discrete parameters. The legends of those figures should also point the reader to a table (or the new figure) describing the acronyms of the geological units.

The figures for the steady-state simulations (e.g., fig 2,3,4,5) have been modified accordingly. However, we did not modify the figures for the transient simulations. This was done because the use of bar-chart plots would have made a comparison hard. An explanation has been added to the paper.

* Across the whole manuscript, the main text should be more self-contained in the sense that description and results of the figures should appear explicitly in the text as well. This is currently done for figs.1,2,7,11 but not the others. The text only mentions that "the results are shown in fig." 3 (L.187), 4(L.200), 10 (L.283) and makes implicit references to figs.5,6,8,9,13,14,15. I don't think fig.12 is even referenced in the text.

Explicit references and descriptions have been added for all figures.

Specific comments:

* A figure is missing to display the geological model, so that the reader can understand better the (implicit) links between the geological units' names (i.e ages) and depth, which is an important aspect of the results. I understand and agree with the approach of not focusing too much on the geologic model (which is quite irrelevant for this study) but a minimum must be mentioned including the number of units, which could maybe be listed in some sort of order of depth (of the centre of mass?) as this is the most relevant aspect.

Please refer also to our answer to a similar comment by reviewer#1. We have added all relevant information as Appendix to the revised manuscript.

* Please check equations 1 and 2 which seem to have a few problems and confirm these typos don't affect any of the results
|— Eq. 1, elements to check/correct:
|—— Minus sign for the diffusion term
|—— $l\_ref^2$ in the source term
|— Eq. 2:
|—— Minus sign for the diffusion term
|—— $l\_ref^2$ in $t\_ref$
|—— $S\_ref$ instead of $S\_{(s,ref)}$ in the second term (heat production)
|—— $S\_{(s,ref)}$ seems to include $C\_p$ and is therefore not necessarily the standard definition of "specific radiogenic heat production", so please specify your definition (and adapt the name if needed).

⊢—— missing "npartial" in front of time

Both equations have been corrected.

* The second part of the introduction (L.48-80) needs some touches to improve the reading flow. The mention of all components of the paper could benefit from adverbs and some reordering to emphasise the logic in which the info is introduced: first "why" some work needs to be done, then "what" are the goals of the paper, and finally "how" you're going about it. For all points, the justification of the work should indeed appear before the mention of the elements themselves, instead of afterwards (which weakens the points by following more of a "report" format, e.g. L.57-59 before L.55; L.61-64 before L.60; L.71-73 before L.70). More explicit logical links will help transform the current impression of a listing of elements ("the main goal of this study" L.48, "we will describe and discuss" L.50, "we will demonstrate" L.52, "the aim of the study" L.54, "in this paper we present" L.60, "our case study is" L.74), which currently leaves the task of connecting them to the reader.

We have rephrased the introduction following the main points raised by reviewer#2.

* L.58 mentions that paleoclimate effects on subsurface heat have only been looked at in 1D and provides a good but old reference from 1984. The impact of paleoclimate on deep heat flux is indeed often underestimated, but more recent work should be mentioned as well (and again as a justification for the work beforehand). See for instance (Dentzer et al, 2016, http://doi.org/10.1016/j.geothermics.2016.01.006) and all references therein.

We would like to thank reviewer#2 for the reference which we have added to the revised version of the manuscript.

* Sec. 2.1, a short sentence would be welcome to actually explain/summarise what the Sobol sensitivity analysis and Saltelli sampling routines are.

An explanation about the Sobol sensitivity analysis and the Saltelli sampling routine have been added.

* Sec. 2.3, briefly mention what kind of constraints are used to calibrate that model, which initial conditions are used.

The model description has been adopted accordingly.

* Sec. 2.3, which absolute time period are the relative time steps 0-26ka supposed to represent? (Reader is only learning l.248 that the number 26k results from the reconstructed paleotemperatures available. This info should appear with the first mention of "26k")

An explanation about the time steps have been added.

* L.172, the contributions being "negligible" imply some relative thresholds that are not specified. An extra sentence would be nice to comment on absolute and relative thresholds the authors used for all indices in this study. For instance, are you choosing three(x2) parameters L.190 because you consider 0.1 to be a good threshold?

An explanation of the selection of parameters and the threshold has been added.

* L.174, the number is not 5 but actually 5x2=10 since later examples show that you're not picking geological units but parameters which are not necessarily in the same units. Please check the whole text for consistency (e.g. "three" L.190)

The text has been revised accordingly.

* L.205, this information should appear at the beginning of the section (as mentioned above, justifications should appear before descriptions): from what I understood, you want to end up with a manageable number (arbitrarily 8) of most sensitive parameters overall but it would be too expensive to run a GSA with all parameters at once, so you break down the problem to first identify the most sensitive in the sediments, then in other areas, and then pick those parameters for further study.

The section has been changed and an explanation that the reduced parameter space is required for both the global SA and the surrogate model construction has been added.

* L.212 "higher accuracy than typical temperature measurements" -l what does that mean exactly? You mentioned that you're not solving for temperature but heat, so how does 5e-4 accuracy on a model translate into temperature measurement precision? (Similarly L.222, why 4e-3 rather than 5e-3 to relax by one order of magnitude?).

The reduced model is constructed using a global error bound. This error bound evaluates the difference for the temperatures at every node between the FE and RB solutions. An explanation has been added to the paper.

* Sec. 3.3.2, I only understood what was being simulated in this section after analysing fig.9 (mentioned L.273). Indeed, Fig.7 (mentioned L.251) shows an average initial temperature around -5C and a final temperature around 8C, which I could not instantly reconcile with applying a Dirichlet boundary condition of 1.6C mentioned L.246.

A detailed description about the chosen boundary condition has been added to Section 3.3.1.

* Sec. 3.3.2, I also don't get the point of fitting the average temperature with a 4th order polynomial. Why not use directly the average temperature itself discretised at your transient time step? Why do you need a smoother version? As for the smoothing, it's impossible to judge a fitting quality without any mention the metric used to assess the impact. I agree (L.277) that the 5th order polynomial doesn't significantly improve the fit compared to the 4th order visually, yet the fit remains rather poor in my eyes (rough estimate of max(deltaT)_3C) and the selection of the best fitting function is a moot point without specifying both the metric to assess the fit and the cost of using a higher order polynomial/smoother fit.

We added the assessment criterion for the fits. Unfortunately, with the current setup of the RB method we require an affine decomposable problem. Therefore, we require at least a piecewise linear function for the upper boundary condition. An explanation has been added to the paper.

* L.280, what is this "scaling factor" and what is it applying to? (I can only start guessing after seeing fig.10, describe it explicitly in the text.)

The scaling factor is used to consider uncertainties of the paleoclimate data. An explanation has been added to the paper.

* L.285, the mention of "glaciation times" comes out of the blue and should be introduced. We added an introduction to the term.

* Sec 3.3.3 looks a bit odd at first sight as it seems to draw an opposite conclusion to the paper itself, with the transient boundary conditions adding no value over constant ones (the main properties showing "no significant changes" L.292 and the others being "insignificant" L.293). Please manage the delivery of this message.
A delivery message has been added to the end of the Section.

* L.416-419, I don't quite get the need to deduce the obvious, that heat moves upwards in this setting, nor the less obvious conclusion of why sediments at the uppermost part have therefore a more prominent influence.
In this paragraph we discuss the relative role of the different geological compartments (based on a gross differentiation between sediments and crustal and mantle domains) on the short to long(er) period effects. While it is true that in our system heat propagates upwards is an obvious conclusion (it could not be differently given the Earth's energy budget), this last statement has an important role in helping discriminating the physical reason why in our sensitivity study the sediments (upper layers) are more prominent over a relative short time period. Indeed, considering heat diffusion as the only energy transport process provides with a diffusive time scale for thermal effects to propagate, which scales as the square root of the system diffusivity over a unit of length scale. Therefore, for a thermal signal in the crust to exert an influence at the level of the sedimentary cover would require a time window of the order of its diffusive time scale times the cumulative thickness through which the signal propagates. A similar reasoning applies for a surface thermal signal to interact with the deeper domains. Based on this physical premise, it is then easy to explain why the sedimentary layers has the most prominent influence if we restrict our analysis to the short-term period. The same is true if we consider time scale of radioactive decay in the crust

which would require additional time to propagate at shallower level given tested values of production rates of crystalline rocks.

* L.443-446, this information would be better suited in section 3 to justify the approach when presenting it.

The sentence has been moved to Section 3.

* Sec. 5, the conclusion needs some polishing. It looks a bit too much like a series of collated dot points with a succession of short sentences (e.g. last paragraph). The paragraph breakdown is awkward with two of them containing a single sentence (L.457, 464). Emphasise more the causal relationships by introducing some segways or logical links, and please amend the abrupt finish to leave the readers on a more impactful last sentence.

The conclusion has been modified accordingly.

Technical corrections
* L.56 & L.58 "influence on [the calibration of] thermal properties"

Has been corrected.

* Sentence L.70-74 could be easier to read if reordered –ı "In this study, we make use of the RB method … since it allows" (l.71) "the retrieval of the entire state variable (i.e. temperature)" (l.73), "in contrast to other statistical methods … " (l.71-73)

Has been addressed.

* L.96 add missing words: [It is] "worth mentioning … "

Has been corrected.

* The last two sentences of sec. 2.1 (L.108-111) should come L.93, after the mention of Sobol GSA (L.89-93) but before the description of the cost function (L.94-108)

Has been corrected.

* (For Eq.(1) and (2), you might want to add a note to point out that the Laplace operator applies with respect to the normalised space. I can see why you wrote those equations this way, to avoid defining all dimensionless variables and parameters, but since you're only using symbols carrying physical dimensions the Laplace operator is slightly misleading, strictly speaking. This pedantic comment is optional.)

A note has been added at the end of Eq. 2.

* Legend of fig.1, not clear (at this point) if the times (0, 13, 26ka) refer to absolute dates (in which case they would be better displayed in inverse order) or are in chronological from an unspecified reference for 0ka.

The legend has been modified to explain the times.

* L.150,151 What are T31 and GR30? Specify a bit the nature of those models and/or add references.

Explanations and references have been added.

* L.157, mentioning the depth of the LAB would be informative.

We have provided a map of the topography of this isothermal and chemical boundary as a supplementary figure (together with all other main geological boundaries of interest) in the Appendix to the revised version of the manuscript.

* Remove comma L.158 "throughout, the entire paper"

Has been removed.

* L.174, why picking 5 parameters (rather than 4 or 6)?

We chose five parameters because of our threshold of 0.1. An explanation has been added to the paper.

* Remove comma L.208 "the investigation[s] carried out so far, have enabled.."

Has been removed.

* Legend of fig.2: Missing mention of the horizontal black line (separation of radiogenic heat and thermal conductivity parameters?) and two boxes (first five) in the figure legend.

The meaning of the boxes and the horizontal (now vertical) black line has been added.

* L.224 "However, with a significantly lower computational cost" (sentence segment, no Verb)

The sentence has been corrected.

* L.224-227: a bit confusing, please rephrase with something along the lines of "Despite potentially introducing additional error sources with a relaxed tolerance, this accuracy drop can actually be considered insignificant. Indeed, sensitivity analyses are based on … .Since all simulations are … see Fig.5."

The sentences have been rephrased for clarification.

* L.230: is "however" the correct logical link?

The logical link has been corrected.

* Sentence L.239-241 As an introduction sentence to the section, keep it at present tense, not past/conditional tense ("having been able" … "could")

The sentence has been corrected.

* Fig.6 branch2 should mention "with paleoclimate"

The figure has been modified accordingly.

* L.251 "Fig. 7 compares the sensitivities of the thermal properties for the steady-state and transient system [with the selected initial and boundary conditions]". One would indeed expect the results of fig.7 to vary with different initial conditions and/or transient boundary conditions.

The sentence has been corrected.

* L.267 "ar[i]sing"

Has been corrected.

* L.275, Eq(3), add something like T_top (t)=âŃŕ to make it a proper equation

Has been addressed.

* L.305 "a second discussion point [is]"

Has been corrected.

* Fig.12, the collage is appropriate, but all fonts need to be slightly increased accordingly

The font size has been increased.

* L.319-320: "very similar" and "showing some major changes" are puzzling/contradictory in the same sentence

The sentence has been reformulated for clarification.

* L.343 not sure I fully understand the wording "combination of the volumetric contributions of the individual layers and their thermal properties". Thermal conductivity and radiogenic heat production are both volumetric properties. What does "and their" imply rather than " ' " ("individual layers' thermal properties")?

The sentence has been changed accordingly.

* L.355 "as apparent by the [insignificant] difference between …"

Has been corrected.

* L. 358: not sure why you put "base" between inverted commas( ?)

We removed the inverted commas.

* L.398, the wording "similar but not identical" doesn't do justice to the importance of this difference.

We provided a more explicit description.

* L.402 "Fig. 6 branch 1 [b]"

Has been corrected.

* L.429 "a[n] additional"

Has been corrected.

* L.440 "Only the consideration of these variations could enable us ..."

Has been corrected.

* L.448 "w[h]ere"

Has been corrected.

* L.450 "fourth" <- cardinal number needed (not ordinal)

Has been corrected.

* L.455: might want to rephrase "since the temperature diffuses over time towards the bottom of the model" as the heat moves upwards and the cold top Dirichlet boundary condition leads to a perceived propagation of a cold front downwards ...

Has been reformulated.

* L.474, reformulate sentence "Using the finite element method the here presented analyses computationally prohibitive, only the utilization of a surrogate model allows the execution of these analyses."

Has been removed since it follows in the next paragraph again.

---

## Author Response (AR2)

This is a points by point review to the comments from both reviewers to the manuscript entitled „Effects of Transient Processes for Thermal Simulations of the Central European Basin" by Degen and Cacace.

Reviewer #1 (anonymous)'s comments:
I reviewed the revised version of the manuscript of Degen & Cacace titled "Effects of Transient Processes for Thermal Simulations of the Central European Basin". The authors addressed all the principal remarks raised in my previous revision. In this revised version, I found minor typos or language inconsistencies that should be checked.

*We would like to thanks the anonymous reviewer for his comments being glad he/she found our answers to his/her previous round of review sound and satisfactory. We have addresses all technical points (grammar and typos) as requested in this second iteration (listed below).*

Here the list referred to the lines of the revised manuscript:

1. L. 28: the second point is not clear considering the introductive sentence "which evolved through geological times". I think that a preposition is missing.

2.  Similarly, the third point could be modified. "lastly" is generally not used in lists since "and" is also present at the end of the second point.

3.  3. L. 65-68: I suggest to rephrase this sentence that has a few subordinate clauses and digressions. This would improve its readability.

4.  4. L. 177: change the reference as "(Maystrenko et al., 2013 and references therein)".

5.  5. L. 286: "and which impact such corrections have on" it seems that a part of the sentence is missing. Please check it.

6.  6. L. 336: probably it is better to divide the sentences with "and". In addition, the dot is missing at the end of the sentence (l.337).

7.  7. L. 337: Fig. 9b instead of Fig. 7b. In addition, ")" is missing.

8.  8. L. 363: the "in" at the beginning of the sentence should be capitalized.

Reviewer #2 (Dr. Thomas Poulet)'s comments:
I am satisfied with the content of all modifications of this revised manuscript and would therefore recommend publication after some technical corrections (that can be assessed by the editor directly).

*We would like to thank Dr. Thomas Poulet for his comments that helped increasing the readability and scientific soundness of our research. We have corrected all minor technical points raised by the reviewer in this second round (those are listed below).*

1. Firstly, the equations still need fixing. If the Laplace operator acts on the normalised space, then there are still some extra L_ref^2 terms in equations 1 and 2. Only those in the first term of both equations should remain. The other three instances should disappear.
Thank your for pointing this out. Indeed, the formulas were wrongly written in the manuscript, and we ahve corrected those accordingly.

2. Secondly, there's a point I hadn't noticed the first time. The maps of fig.1 at 0ka and 26ka look strangely similar. Is this really true or was the same subfigure copy-pasted by mistake?
Thank you for noticing. There was indeed an error in the plot, which has now been corrected.

All language remarks have been addressed in the document.
Lastly, the English remains a bit difficult to follow at times and will hopefully be polished a little bit by the editorial team. Here are a few points regarding the modified text:
• The bullet points listed l.27-29 should make sense individually when read after l.26. A possible modification is "… which evolved through geological times due to: (1) varying thermal loading conditions …, (2) the amount of heat …, and, (3) lastly, the (time-varying) surficial…"
• Same comment for the sentence l.79-81. "…from the fact that the CEBS (i) represents…, and (ii) is an area of interest …"
• Phrase segment l.100 "as higher first-order than total-order contributions" is not clear
• L.165: unclear sentence "For the ocean resolution three degree has been used"
• L.277 change "variable in time" to "time-varying"
• L.283 "no" -> "i.e. without"
• L.284 "Only after we have been able to quantify" -> "After having quantified"